# Characterizing cancer metabolism from bulk and single-cell RNA-seq data using METAFlux

Yuefan Huang[1,2], Vakul Mohanty[1], Merve Dede [1], Kyle Tsai[1], May Daher [3], Li Li[3], Katayoun Rezvani [3] & Ken Chen [1] ✉

Cells often alter metabolic strategies under nutrient-deprived conditions to support their survival and growth. Characterizing metabolic reprogramming in the tumor microenvironment (TME) is of emerging importance in cancer research and patient care. However, recent technologies only measure a subset of metabolites and cannot provide in situ measurements. Computational methods such as flux balance analysis (FBA) have been developed to estimate metabolic flux from bulk RNA-seq data and can potentially be extended to single-cell RNA-seq (scRNA-seq) data. However, it is unclear how reliable current methods are, particularly in TME characterization. Here, we present a computational framework METAFlux (METAbolic Flux balance analysis) to infer metabolic fluxes from bulk or single-cell transcriptomic data. Large-scale experiments using cell-lines, the cancer genome atlas (TCGA), and scRNA-seq data obtained from diverse cancer and immunotherapeutic contexts, including CAR-NK cell therapy, have validated METAFlux's capability to characterize metabolic heterogeneity and metabolic interaction amongst cell types.

Metabolism is essential for proper cellular function. Cancer cells harboring aberrant genetic alterations such as amino acid substitutions and copy number alterations, often exhibit distinct metabolic programs from normal cells[1,2], an established hallmark of cancer[3]. Clinical studies have demonstrated that metabolism is associated with patient outcomes and that specific metabolic phenotypes could present vulnerabilities to cancer treatment[4]. For example, upregulation of fatty acid oxidation (FAO) has been shown to fuel acute myeloid leukemia (AML) venetoclax with azacytidine (ven/aza) resistance and inhibition of FAO may restore the efficacy of ven/aza treatment[5]. A recent study shows that dysregulated propionate metabolism increases the metastatic potential for breast and lung cancer[6]. Therefore, understanding how metabolic dysregulation promotes cancer is key to treating it.

Over the past two decades, technological innovations have allowed for detailed characterization of metabolic alterations. These established techniques, such as metabolomics, stable isotope tracing, and XF Extracellular Flux Analyzer, have facilitated discoveries in

metabolism from different perspectives. LC/MS (liquid chromatography/mass spectrometry) based metabolomics is a powerful tool for measuring concentration of metabolites and is often the first choice of metabolic profiling[7]. Recent advancement in MALDI-MS has also enabled metabolomics profiling with around 100 molecules detected at 5–10 um resolution[8,9]. However, the process of metabolite identification for metabolomics is still low throughput, requiring considerable time and effort to analyze the datasets[10]. Moreover, reproducing results is challenging due to complexity of the experiments and a lack of methodology standardization[11]. In addition, metabolomics only provides static snapshots, missing dynamic profiles of metabolite trafficking, velocities of metabolic reactions (a.k.a. fluxes), etc., which are critical to understanding mechanisms of metabolic regulation.

Consequently, metabolic flux techniques, for instance, [13]C metabolic flux analysis ([13]C-MFA) and Seahorse Extracellular Flux (XF) analyzer, have become widely used in metabolic research[10]. [13]C-MFA is the current gold standard in measuring intracellular fluxes of central

[1]Department of Bioinformatics and Computational Biology, The University of Texas MD Anderson Cancer Center, Houston, TX 77030, USA. [2]Department of Biostatistics & Data Science, School of Public Health, The University of Texas Health Science Center at Houston (UTHealth), Houston, TX 77030, USA. [3]Department of Stem Cell Transplantation and Cellular Therapy, The University of Texas MD Anderson Cancer Center, Houston, TX 77030, USA. ✉e-mail: kchen3@mdanderson.org

carbon metabolism, while the Seahorse Extracellular Flux (XF) analyzer is the benchmark for assessing cells' extracellular bioenergetic state. Although [13]C-MFA offers valuable insights, it has limited usage in deriving large metabolic networks[12]. Seahorse Extracellular Flux (XF) analyzer can measure OCR (Oxygen consumption rate: an indicator of mitochondrial respiration) and ECAR (extracellular acidification rate: an indicator of glycolysis) of living cells simultaneously in real-time[13]. Even though the Seahorse platform provides valuable insight into the functional status of cells, fluxes of other metabolites are not measured. For in vivo metabolic assessment, approaches such as positron emission tomography (PET) are used. These include techniques like [18]F-fluorodeoxyglucose (FDG) PET and [18]F-Glutamine[14]. In addition, in vivo stable isotope tracing has emerged as a novel approach, utilizing stable isotope-labeled nutrients to investigate metabolic activity within intact tumors[15]. Notably, these methods are limited to probing specific subsets of metabolic reactions[14].

Beyond limited scales, substantial experimental challenges exist in studying cancer metabolism in culture. It is difficult to accurately mimic a complex metabolic environment[16], consisting of a dynamic mixture of malignant and non-malignant cells. As a result, traditional cell culture nutrient milieu does not accurately resemble human physiological nutrient environment[17,18].

The advent of RNA-seq and scRNA-seq provided an opportunity to systemically interrogate the transcriptomic profiles of biospecimens at tissue and cellular resolution[19,20]. Given the challenges of metabolic profiling, transcriptomic analyses have been utilized as a surrogate to reveal metabolic reprogramming and vulnerabilities in tumors[21–24]. Those studies often employ statistical methods that score mRNA expression levels of genes in a predefined (e.g., KEGG) metabolic pathway. Several scoring methods exist, for example, ssGSEA[25], Seurat AddModuleScore[26], AUCell[27], singscore[28], Z-score, etc. However, these methods cannot be applied at reaction level because they become unstable for metabolic reactions involving a few (usually less than ten) genes[29]. Moreover, they cannot be applied to non-enzymatic reactions. For example, previtamin $D_3$ forms vitamin $D_3$ via spontaneous reaction[30]. More importantly, these methods examine each pathway disjointly, ignoring the fact that metabolic networks are highly connected and dynamic[31].

To interrogate the entire metabolic circuits, genome-scale metabolic models (GEMs), which encapsulate an organism's stoichiometrically balanced metabolic reactions via gene-protein-reactions (GPR) association[32], have been utilized. One of the most common analytic methods used in GEMs is flux balance analysis (FBA), a well-established constrained optimization method that estimate flow of metabolites in a complex bio-system[33]. It maximizes or minimizes the flux of a particular reaction or a linear combination of reactions, under steady-state assumptions and flux bounds constraints. Versions of FBA have been successfully applied in various settings[34–37].

To apply FBA on gene expression data, gene expression levels need to be systematically interpreted in the context of metabolic networks. Previous studies have shown such exercises could lead to rational estimation of metabolic fluxes[38–49]. A few varieties exist, one of which is to define an objective function that includes gene expression levels. For example, Lee et al. characterize the biological objective function based on correlation between fluxes and gene expression levels[43]. Similarly, iMAT uses gene expression levels to dichotomize highly expressed and lowly expressed reactions and finds the flux distribution that best explains gene expression patterns. It maximizes the number of reactions classified as highly expressed and minimizes the number of reactions classified as lowly expressed[38]. Promising efforts have been made to predict COVID-19 metabolic targets and changes using iMAT on scRNA-seq data[50,51]. Another way to incorporate gene expression data into FBA is to define flux bounds in FBA using expression levels. For example, E-Flux uses transformed gene expression levels as flux upper and lower bounds[45]. Also, Damiani et al.

estimate single-cell fluxomics using modified transformed expression values as constraints[52]. Although studies have demonstrated the utility of combining gene expression and FBA in assessing metabolic states, many do not directly produce unique flux distributions, nor include biologically meaningful constraints that account for nutrient exchange or competition among cell types in the TME. Moreover, systematic validation of in silico estimation is largely lacking.

In this work, we introduce METAFlux, a computational tool that predicts cancer metabolic fluxes from bulk RNA-seq and scRNA-seq data to address these analytic gaps. METAFlux is capable of characterizing the entire metabolic circuits and output non-degenerative fluxes using cancer gene expression data in a nutrient-aware manner. For scRNA-seq data, METAFlux additionally examines metabolic heterogeneity and interactions amongst cell types in TME. We evaluate METAFlux prediction accuracy using the matched flux data generated on the NCI-60 cell lines and find a substantial improvement over existing approaches. We further examine METAFlux on scRNA-seq data obtained from an in vivo Raji-NK cell co-culturing model and observe high consistency between the predicted and experimental (i.e., Seahorse extracellular) flux measurements. METAFlux only requires gene expression data as input and is customized to fit binary experimental conditions (nutrient presence vs absence). METAFlux can predict 13,082 reaction flux scores for each bulk sample, and for each single cell data, it can predict (13,082 × number of cell-type/ cluster + 1648) reaction flux scores. Because of the wide availability of RNA-seq and scRNA-seq data, METAFlux can dramatically improve our understanding of metabolism in disease samples at both tissue and cell-type levels. Results produced by METAFlux can serve as a resource to identify metabolic targets for precision medicine. The open-source implementation is available at https://github.com/KChen-lab/METAFlux.

## Results

### Modeling metabolism using transcriptomes

METAFlux utilizes Human1, a genome-scale metabolic model (GEM) that encodes the mechanistic relationships between genes, metabolites, and reactions in a human cell. Human1 integrates the Recon, iHSA, and HMR models[53]. It contains 13,082 reactions and 8378 metabolites[53]. We choose Human1 because it shows a considerable improvement over other GEMs in terms of stoichiometric consistency, percentages of mass, and charge-balanced reactions[53].

For each sample in a bulk dataset, we first compute a metabolic reaction activity score (MRAS) for each reaction, which describes the reaction activity as a function of the associated gene expression levels (Fig. 1a, Methods). Subsequently, we define a nutrient environment profile, which includes a binary list of metabolites available for uptake (Methods). We hypothesize that tumors proliferate rapidly; thus, the new human biomass pseudo-reaction, which constructs a generic human cell's nutrient demand and composition, should be optimized[53]. We next apply convex quadratic programming (QP) that simultaneously optimizes the biomass objective and minimizes the sum of fluxes' squares, similar to a previous approach[54] (Methods). We also propose a workflow for single-cell data, where the whole TME is modeled as one community to account for metabolic interaction between the groups with the whole community biomass optimized, since we believe cell groups strongly influence one another in TMEs (Fig. 1b, Methods)[55].

### Benchmarking the performance of METAFlux using experimental data

We benchmarked METAFlux using the NCI-60 RNA-seq data and matched metabolite flux data[53,56]. We selected 11 cell lines for evaluation, as other cell lines had nutrient depletion that could affect reliability of flux profiling[53,56–58]. Each cell line had 26 experimentally measured metabolite fluxes and one biomass flux.

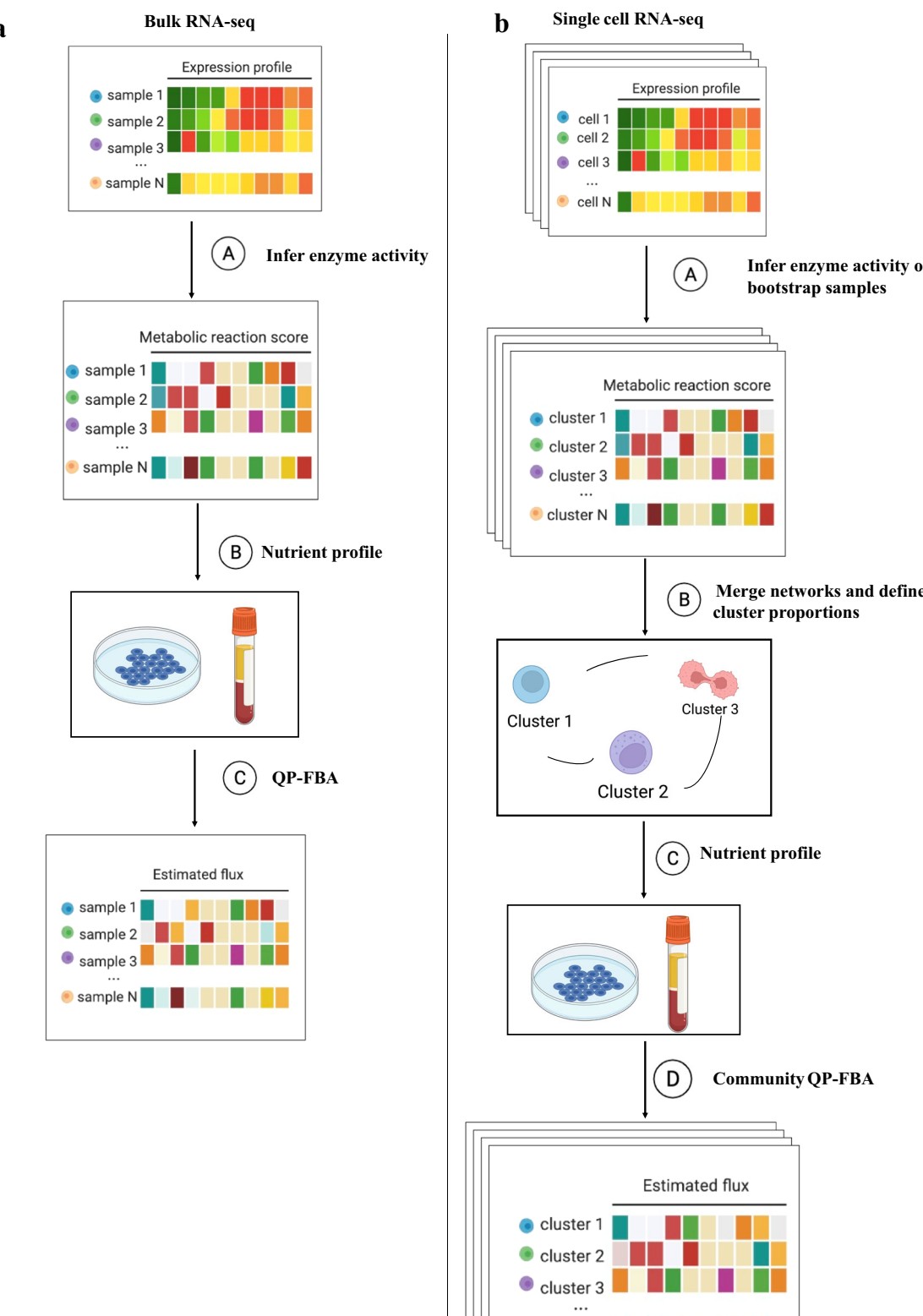

**Fig. 1 | The workflow of METAFlux. a** The workflow of METAFlux in bulk RNA-seq setting. In step A, metabolic reaction activity scores (MRAS) are estimated from RNA-seq data. In step B, a nutrient profile is defined so only certain nutrients can be uptaken. In step C, quadratic programming-based FBA (flux balance analysis) is performed to estimate metabolic fluxes for each sample. The figure was created with BioRender.com. **b** The workflow of METAFlux in single-cell RNA-seq setting. In step A, metabolic reaction activity scores (MRAS) are estimated for each stratified bootstrap sampled single-cell dataset. In step B, metabolic networks for different clusters are merged to form one community, and proportions of clusters should be defined during this step. In step C, nutrient profile is defined so only specific metabolites can be uptaken by TME. In step D, community-based quadratic programming FBA is constructed to estimate per cell average metabolic fluxes for each cluster and total average metabolic fluxes for overall TME. The figure was created with BioRender.com.

We ran METAFlux based on *cell line culture* medium composition (Supplemental Data File S1 and Methods). Meanwhile, we compared METAFlux results with those generated by a state-of-art pipeline, *ecGEMs*, which predicts flux based on a cell-type specific GEM with reactions constrained by the gene expression levels, enzyme abundance, and kinetics[53,59,60]. The results are comparable with those of METAFlux because the programs optimize the same scoring objective, i.e., biomass, under identical medium compositions.

Overall, METAFlux achieved markedly higher Spearman correlation with the ground truth than did ecGEMs (Spearman coefficients ρ:

0.74 vs 0.45). In all the 11 cell lines, METAFlux performed better (Fig. 2a). Across 26 metabolites, METAFlux achieved higher correlation in 16 of the cases (Fig. 2b). To evaluate directionality of the flux prediction, we categorized metabolic fluxes as 'no flux' (flux equals to zero), 'uptake' (flux smaller than zero), and 'excrete' (flux greater than zero). Except for 'choline,' METAFlux achieved higher accuracy, defined as the percentage of categorical match with the ground truth, for seven metabolites and tied with ecGEMs for the other 18 (Fig. 2c). Taken together, these results indicate that METAFlux outperforms ecGEMs on predicting metabolic fluxes from cell line RNA-seq data.

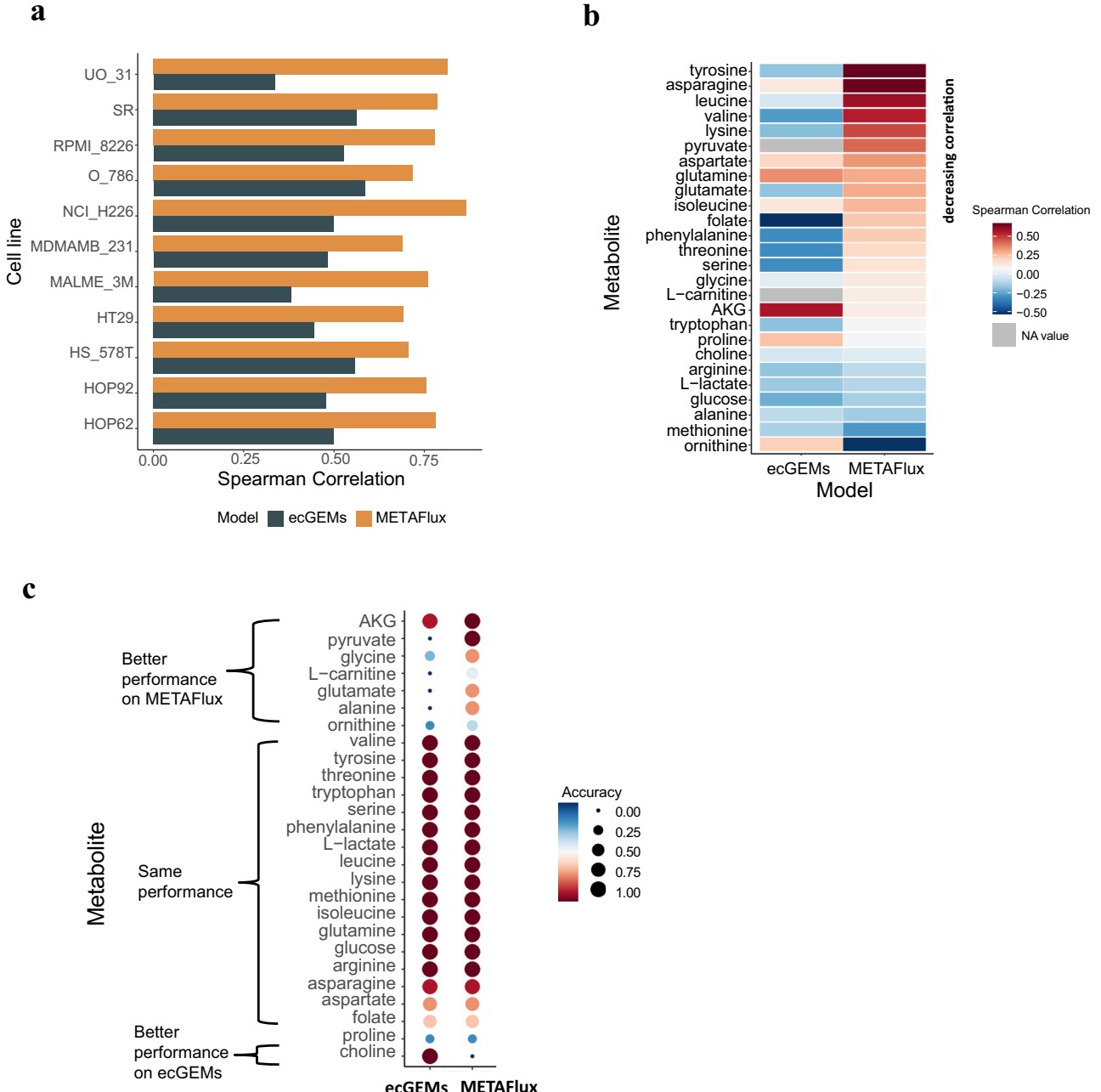

**Fig. 2 | Benchmark results of METAFlux on NCI-60 cell lines. a** A Spearman correlation bar plot across each cell line for METAFlux and ecGEMs. The spearman correlations between predicted fluxes and experimental fluxes were calculated for 11 cell lines. **b** A Spearman correlation heatmap across each metabolite for METAFlux and ecGEMs. The spearman correlations between predicted fluxes and experimental fluxes were calculated for 26 metabolites. For metabolites like

L-carnitine and pyruvate, ecGEMs predicted their fluxes to be zero. Therefore, the correlations of these metabolites to ground truth were not calculated. **c** Uptake and secretion direction accuracies of 26 metabolites for METAFlux and ecGEMs. The accuracies were defined by the ratio of the number of direction-aligned fluxes to the total number of fluxes for each metabolite. The accuracies are shown by color and dot size.

In our study, we recognize the constraints of employing the NCI-60 panel for examining glucose uptake, given that correlations for other metabolites appear more consistent. This discrepancy may be because the experimental model may not fully capture the complexity of glucose metabolism, and there is a difference in nutrient and metabolic requirements between cell lines in controlled environments and tumors in the tumor microenvironment. To further validate the performance of METAFlux in a more physiologically relevant context, we assessed its ability to predict glucose uptake in a cohort of 84 triple-negative breast cancers (TNBCs) using FDG-PET data[61]. These TNBC samples had matched gene expression and Standardized Uptake Value (SUV) data, with the SUV representing a measure of glucose uptake derived from the radioactivity concentration in tissue and the injected dose of radioactivity per kilogram of the patient's weight. We analyzed these samples under a *human blood* profile medium (Supplementary Data File S1) and found a statistically significant Spearman correlation of 0.31 (*P* value = 0.003) between the predicted glucose uptake and the measured SUV (Supplementary Data File S2 and Supplementary Fig. 1a). These results underscore the potential of METAFlux to accurately predict glucose uptake in larger cohorts and settings more representative of the complex tumor microenvironment, highlighting its utility for studying metabolic fluxes in actual tumors.

We also assessed METAFlux's ability to predict glucose uptake in low-perfusion, nutrient-poor tumors by analyzing a dataset of pancreatic cancers[62] (*N* = 8) using a *human blood* profile (Supplemental Data File S1 and Supplementary Fig. 1b). The dataset contains matched gene expression and Standardized Uptake Values (SUV) derived from FDG-PET. We included all 64 metabolites from the list, because the final predicted rates should reflect the limiting condition within the tumor microenvironment, as fluxes are collectively constrained by gene expression and steady-state assumption.

## Biologically meaningful medium resulted in better prediction

To estimate the effect of medium constraints on modeling the NCI-60 data, we performed control experiments that did not impose any medium constraints in running METAFlux, allowing free uptake or excretion of all metabolites. Under such a setting, METAFlux only achieved an overall Spearman correlation of $\rho = 0.15$. Spearman correlations of individual metabolites or cell lines also dropped significantly (Supplementary Fig. 1c, d).

We further estimated the statistical significance of the accuracies calculated by METAFlux under real medium conditions against those under random conditions (Methods). Zero P-values were obtained for both the overall Spearman correlation and the directionality accuracy, indicating that the results were unlikely to have been obtained by chance (Supplementary Fig. 1e, f). Taken together, these results indicate that METAFlux has achieved biologically meaningful modeling of cancer cell-line metabolism in vitro.

## METAFlux revealed a metabolic subtype of lung cancer in TCGA

To examine METAFlux in real patient data, we applied it to TCGA LUAD (lung adenocarcinoma) and TCGA LUSC (lung squamous cell carcinoma) data using *human blood* profile as medium constraints (Methods). We examined glucose uptake flux (GUF) predicted by METAFlux from the LUSC and the LUAD RNA-seq data, respectively. The results indicated that LUSC tumors had a higher GUF than did LUAD tumors, which is consistent with the [18]F-FDG PET-CT (18F-fluorodeoxyglucose positron emission tomography) scan results (Fig. 3a and Supplemental Note) performed in an independent study[63]. In addition, we found that the predicted GUF is highly correlated (Spearman correlation $\rho = 0.65$) with proliferation signature scores obtained in an independent study[64] (Supplementary Fig. 2a), consistent with another independent finding that [18]F-FDG PET is closely correlated with proliferation index[65].

Clustering METAFlux metabolic flux profiles revealed two clusters of LUAD and LUSC samples (Fig. 3b and Supplementary Methods). Cluster 1 consists primarily of LUAD samples, while cluster 0 is a mixture of LUAD and LUSC samples (Supplementary Fig. 2b). The LUAD samples in cluster 0 can be called 'LUSC-like' LUADs, due to similarities in the metabolic phenotype. The LUAD samples in cluster 1 (LUAD1) had significantly lower glucose uptake than 'LUSC-like LUAD" (Wilcoxon Rank Sum *P* values < 0.0001), implying that LUSC-like LUAD tumors are metabolically more active (Fig. 3c). Survival analysis revealed that LUAD1 tumors had significantly better survival outcomes (*P* value = 0.0084) than 'LUSC-like' LUADs (Fig. 3d and Supplementary Fig. 2c, Supplementary Methods). LUAD1 remained significantly associated with reduced mortality risk compared to reference group LUSC-like LUAD even after adjusting for demographic and clinical variables (Supplementary Table S1). We then sought to identify the "LUSC-like LUAD" patient group using FDG-PET. We analyzed matched TCGA LUAD FDG-PET metadata containing Standardized Uptake Value (SUV) max values obtained from a previous study[63]. The LUAD FDG-PET patient cohorts included four LUAD1 and nine LUSC-like LUAD patients. Our findings revealed a trend of higher glucose uptake in LUSC-like LUAD patients (*T* test *p* = 0.086) (Supplementary Fig. 2d). Although not strictly significant due to the small sample size, this trend aligns with our METAFlux analysis. Gene set enrichment analysis suggested that bile acid metabolism was upregulated in cluster 0 (NES = 1.59, BH adj. *P* value = 0.04), which is consistent with an earlier study, indicating an association between bile acid metabolism upregulation and worse prognosis in LUAD[66].

Notably, such a result cannot be obtained from the metabolic gene expression data, without applying METAFlux assessment (Supplementary Fig. 2e–h and Supplementary Methods). This could be attributed to integration of external knowledge about metabolic pathway networks and dependence among genes. These results demonstrated a distinct utility of METAFlux in discovering tumor subtypes from large transcriptomically profiled cancer patient cohorts.

## METAFlux can capture hypoxia-associated metabolic reprogramming

Next, we sought to address the significance of hypoxia, a condition characterized by limited oxygen supply, in tumor microenvironments. We explored whether METAFlux could capture hypoxia-driven metabolic changes through gene expression constraints. Using TCGA pan-cancer data, we examined metabolic adaptations under hypoxic conditions, which typically involve increased glucose uptake and lactate production in oxygen-deprived cells[67]. We performed additional analysis on TCGA pan-cancer data and retrieved the hypoxia signature score from a previous study[68]. mRNA-based hypoxia scores were available for 676 tumors. The scores were calculated by assigning a +1 score to patients with mRNA abundance in the top 50% for each gene in the winter hypoxia signature and a −1 score to those in the bottom 50%. This process was repeated for every gene in the signature, generating a hypoxia score for each patient[68,69]. Higher scores indicate hypoxia, and lower scores indicate normoixa.

Subsequently, we regressed METAFlux features against the hypoxia scores: *lm(Hypoxia scores ~ Oxygen_Uptake + Glucose_ uptake + Lactate_secretion)* (Supplementary Table S2). Our results showed that oxygen uptake was negatively associated with hypoxia score (Estimate = −0.19, CI = −0.27 to −0.12, *p* < 0.001), while glucose uptake was positively associated with hypoxia score (Estimate = 0.32, CI = 0.25–0.40, *p* < 0.001). Lactate secretion was also positively associated with hypoxia score (Estimate = 0.09, CI = 0.02–0.17, *p* = 0.014). These results align with hypoxia-mediated metabolic reprogramming and demonstrate METAFlux's capacity to accurately capture hypoxia-associated metabolic adaptations.

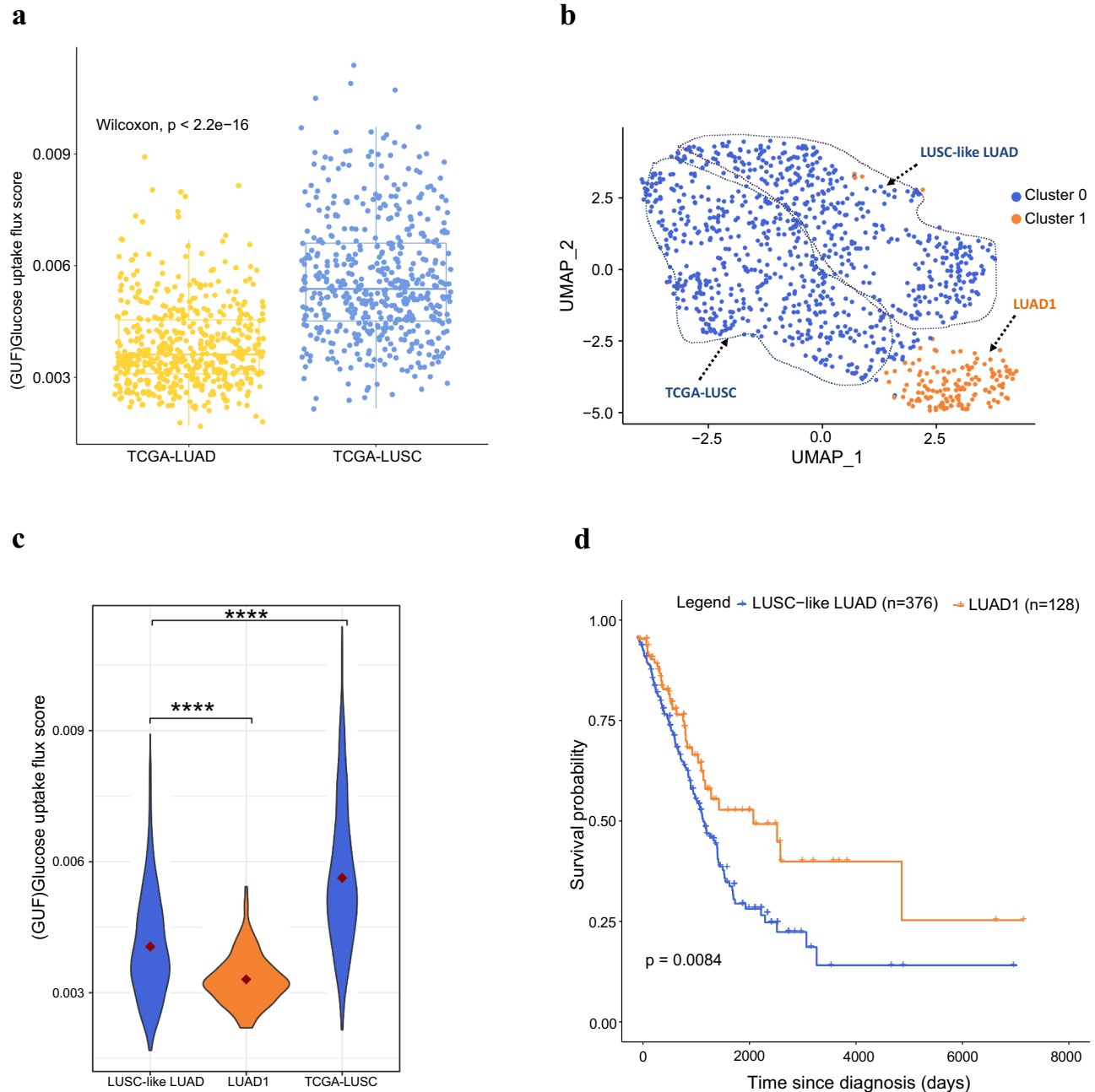

**Fig. 3 | Metabolic characterization of TCGA LUAD and LUSC using METAFlux.**
**a** A boxplot of predicted glucose uptake activity in LUAD and LUSC. The box represents the interquartile range (IQR), the line within shows the median, and the whiskers extend up to 1.5IQR. Min and max values correspond to the smallest and largest mean predicted glucose flux. *P*-values were calculated using the two-sided Wilcoxon Rank Sum test. **b** A UMAP of TCGA LUAD and LUSC samples using predicted metabolic fluxes. Two clusters were identified within those samples. Cluster 1 was enriched with LUADs, while cluster 0 contained both LUADs and LUSCs. **c** A violin plot of glucose uptake for LUSC-like LUAD, LUAD1, and LUSC. *P* values were calculated by the two-sided Wilcoxon Rank Sum test. *P*-values are denoted as follows: ns ($p > 0.05$), *($p \le 0.05$), **($p \le 0.01$), ***($p \le 0.001$), ****($p \le 0.0001$). *P*-value = $7.0 \times 10^{-8}$ for LUSC-like LUAD and LUAD1 comparison. *P*-value = $2.1 \times 10^{-5}$ for LUSC-like LUAD and TCGA-LUSC comparison. The mean glucose uptake for each cluster was denoted as a red circle. **d** Overall Kaplan-Meier survival curves for cluster 0 (LUSC-like LUAD) and cluster 1 (LUAD1) in TCGA LUADs. Clusters were generated using metabolic fluxes.

## METAFlux identified different architypes of metabolism in lung cancer bulk sorted data

Different cell-types in a TME can have distinct metabolic programs. Such heterogeneity can be revealed by applying METAFlux on sorted RNA-seq data. We applied METAFlux in a community-based setting (Methods) on data acquired from primary lung cancer patients[70], flow-sorted into immune cells (CD45+ EPCAM−), endothelial cells (CD31+ CD45− EPCAM−), tumor cells (EPCAM+ CD45− CD31−), and fibroblasts (CD10+ EPCAM− CD45− CD31−). After data processing (Supplementary Methods), we had 15 lung adenocarcinoma (LUAD) and nine lung squamous cell carcinoma (LUSC) samples with complete RNA-seq profiles for all four cell types. We used the cell type proportions calculated using CIBERSORT in the original study[70] as input parameters to run METAFlux.

Comparing the resulting predicted total glucose uptake between the LUSCs and the LUADs, trends consistent with the bulk data were

found: LUSCs had higher overall glucose uptake than the LUADs (*P*-value = 0.058, 95% CI of effect size [−1.8149, −0.1191], two-sample *t*-test) (Supplementary Fig. 3a). The LUSC tumor cells appeared to have slightly higher predicted glucose uptake than the LUAD tumor cells on average (*P*-value = 0.222, 95% CI of effect size [−1.3435, 0.2831], two-sample *t*-test) (Fig. 4a), while those in the endothelial and fibroblast cells were similar between the LUSCs and the LUADs (endothelial *P*-value = 0.762, 95% CI of effect size [−0.9300, 0.6677], fibroblasts

*P*-value = 0.803, 95% CI of effect size [−0.9037, 0.6934], two-sample *t*-test). However, a significant difference emerged in immune cells (*P*-value = 0.045, 95% CI of effect size [−1.8626, −0.1579], two-sample *t*-test), indicating that immune cells may not be deprived of glucose in the TMEs of the LUSCs compared with those in the LUADs. Alternatively, it can also suggest that a specific subtype of immune cells may drive glucose uptake in a population of immune cells. Moreover, the metabolic profiles of immune cells, which typically elevate upon activation, could be significantly influenced by variations in their activation states.

To gain insights at cell type level, we multiplied the per cell nutrient uptake flux by corresponding cell type abundance calculated using CIBERSORT (Supplementary Fig. 3b and Supplementary Methods). We observed a variety of heterogeneous affinities for metabolites and distinct metabolic phenotypes across the cell types. Cancer cells have a strong preference for several amino acids (e.g., glutamine, methionine, ornithine, and lysine) and oxygen. Those altered amino acid metabolisms has been recognized as a hallmark of malignancy[71]. While tumor cells appeared to dominate across the TMEs, immune cells also contributed large fractions in citrate and 2-hydroxybutyrate, especially in LUSC. Citrate metabolic pathways, connecting with lipid and glucose metabolism, are known to play crucial roles in regulating immune cell functions[72].

We further revealed a variety of metabolic interaction architypes in the TMEs of both LUAD and LUSC (Fig. 4b–d and Methods) implicating competitions for metabolites, such as glucose, oxygen, glutamine, and other amino acids (Fig. 4b and Supplementary Table S3), as well as cooperation, by which one or more cell types utilize nutrients (e.g., phenylalanine) released by other TME components (Fig. 4c and Supplementary Table S3). Interestingly, we also found a *Release* architype where all TME compartments released a certain nutrient (e.g., lactate) (Fig. 4d). Taken together, our study highlights META-Flux's ability to discover metabolic crosstalk between various TME cellular compartments, which can inform development of systems medicine.

## METAFlux uncovered metabolic heterogeneity from single-cell lung cancer data

To further deconvolute the TME and assess METAFlux, we examined a single-cell LUAD dataset containing seven patients with lymph node metastasis samples (Fig. 5a)[73]. We excluded cell types with an average proportion of less than 5%. A total of 25,536 cells, including B cells, epithelial cells, myeloid cells, and T cells, were used to construct a single-cell metabolism map using METAFlux community setting (Fig. 5a, b). Except for one outlier (EBUS_12), we found that glucose uptake of the TME has a moderate correlation with myeloid infiltration level in the lymph nodes (Fig. 5c). This finding agrees with previous studies that have reported a correlation between myeloid infiltration and FDG avidity[74,75]. We further examined glucose uptake levels of each cell type and uncovered a large variety of inter- and intra-patient heterogeneity (Fig. 5d). Overall, epithelial cells (CD45- cells, including both normal and tumor epithelial) appeared to have lower glucose uptake than do immune cells (CD45+ cells) (*P* value < 0.0001, two-sample *t*-test) on average. Myeloid cells had the highest level of glucose uptake, followed by B cells, T cells, and epithelial cells (Myeloid cell >B cell >T cell >Epithelial cell, *P* value < 0.0001, Cuzick trend test).

Our study also revealed a complex metabolic interplay with cooperation and competition relationships in metastatic LUAD (Fig. 5e and Supplementary Fig. 4a). Serine, a crucial precursor for protein synthesis, nucleic acids, and lipids, was differentially taken up and released by tumor and immune cells. The release of serine by B and T cells could support tumor cell growth, while myeloid cells' serine uptake may compete with tumor cells, indirectly modulating serine availability. Epithelial (tumor) cells exhibited the highest arginine uptake, indicating increased demand for protein synthesis, polyamine

**a**

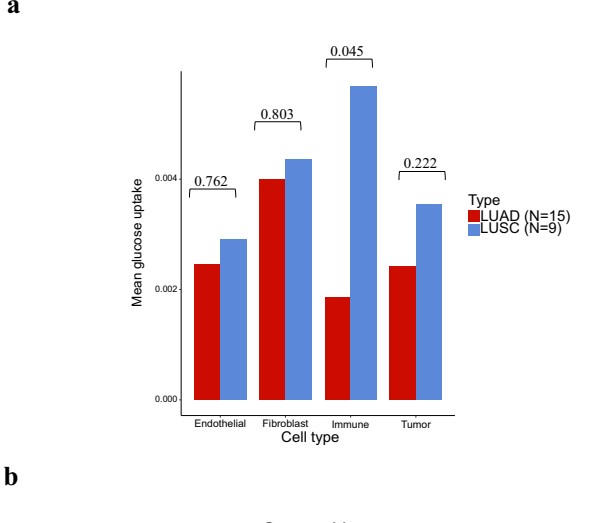

**b**

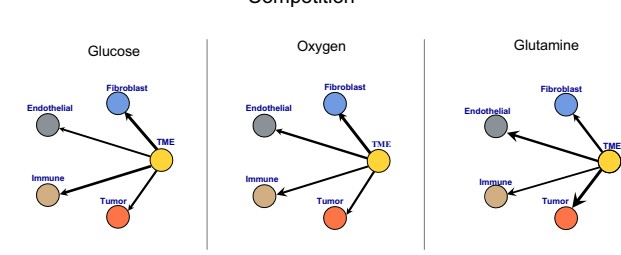

**c**          **d**

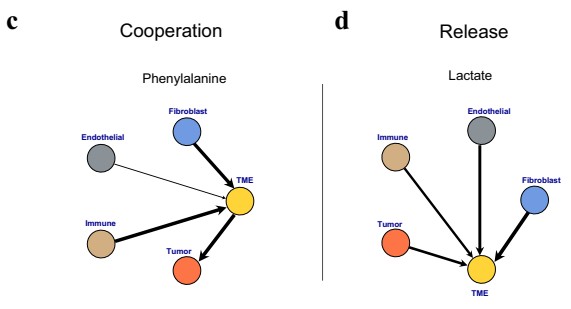

**Fig. 4 | Cell type metabolic fluxes and architypes identified by METAFlux.** **a** Average per-cell glucose uptake fluxes by cell-type in LUAD and LUSC. Two sample *T* test *P*-values are labeled at the top. **b**–**d** Graph-based representation of three different metabolic architypes found in the average profile of LUAD and LUSC. Each node represents one cell type (Immune, endothelial, tumor, or fibroblast) or shared TME. The edge width represents the absolute magnitude of flux, calculated by averaging the per-cell LUAD and LUSC flux. The arrow shows the direction of flux. Arrow coming from cells to TME means the nutrient of interest is released to TME. Arrow coming from TME to cells means the nutrient of interest is absorbed by cells. **b** Competition. The three examples shown are immune, endothelial, tumor and fibroblast cells competing respectively for glucose, oxygen, and glutamine uptake in the TME. **c** Cooperation. The example shows that immune, endothelial, and fibroblast cells released phenylalanine into the TME while tumor cells uptook phenylalanine from the TME. **d** Release. The example shows that lactate was being released into TME by all the cell types.

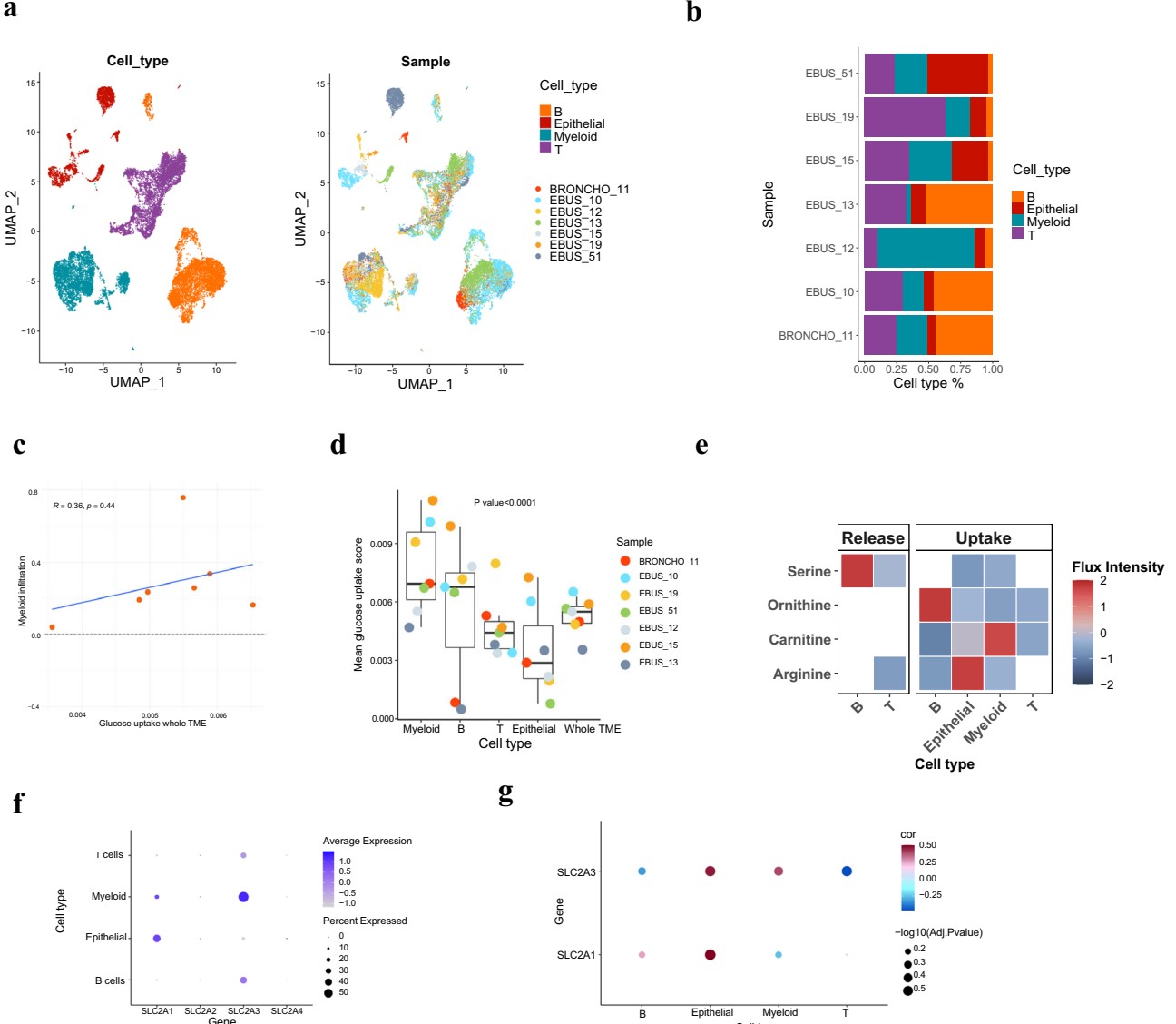

**Fig. 5 | Metabolic characterization of metastatic LUAD from single-cell RNA-seq data.** **a** A UMAP embedding of scRNA-seq data from seven metastatic LUAD patients. **b** A bar plot of cell type percentages in each patient. **c** A scatter plot showing the relationship between mean glucose uptake of the whole TME and myeloid infiltration score. The blue line represents the linear regression line, and the grey band shows a 95% confidence interval. The Spearman correlation coefficient with two-sided P-value is shown on the top left. **d** A box plot of mean glucose uptake flux of each cell type and the whole TME for the seven patients. The box represents the interquartile range (IQR), the line within shows the median, and the whiskers extend up to 1.5IQR. Min and max values correspond to the smallest and largest mean predicted glucose flux. Test: Cuzick trend test. *P*-value = $2.0 \times 10^{-5}$. **e** The release and uptake profile of selected metabolites for each cell type. The color represents the absolute flux intensity. **f** A dot plot showing the mean glucose transporter genes expression levels for each cell type. The dot size represents the percentage of cells in that cell type expressing a given gene, and the color gradient means the average expression value for that cell type. **g** A dot plot showing the Spearman correlations between the observed glucose transporter gene expression levels and the predicted mean glucose uptake flux for a given cell type across seven patients. The color in a dot corresponds to the coefficient of a correlation, and the size represents the FDR-corrected two-sided *P*-value of the correlation.

biosynthesis, and nitric oxide production. All cell types took up ornithine, another polyamine precursor, with B cells showing elevated uptake. Carnitine, essential for fatty acid metabolism, was taken up by all cell types, with the highest uptake in myeloid and epithelial (tumor) cells, suggesting a reliance on fatty acid metabolism and potential competition for resources. Further subpopulation analysis, provided in the Supplementary Note, offers additional insights into these intricate relationships.

We then sought out to compare METAFlux's glucose uptake results with ssGSEA and Seurat AddModuleScore (Supplementary Fig. 4b, c). The ssGSEA results showed a trend of Epithelial cells > Myeloid cells > T cells > B cells (*P* value < 0.0001, Cuzick trend test),

while Seurat AddModuleScore results revealed a trend of Myeloid cells > Epithelial cells > T cells > B cells (*P* value < 0.0001, Cuzick trend test). Correlating glycolysis pathway scores with METAFlux predicted glucose uptake level for seven patients, we found that only Seurat AddModulescores had good consistency with METAFlux for epithelial cells (Supplementary Fig. 4d and Supplementary Table S4). However, they did not show significant positive correlations with predicted glucose uptake levels in other cell types. In summary, METAFlux predicted a different trend in glucose uptake than the traditional pathway scoring methods. This could be attributed to METAFlux's ability to treat different cell types in TME as one community and take interactions among cell types into account.

We further investigated if the predicted glucose uptake could be substantiated by glucose transporter (*GLUT*) gene expression levels. We observed that *GLUT1* (*SLC2A1*) is expressed mainly in epithelial cells and *GLUT3* (*SLC2A3*) expressed mostly in immune cells, particularly myeloid cells (Fig. 5f). Thus, single-cell gene expression data suggests that different cell types leverage different glucose transporters in the TME. We then correlated METAFlux-predicted glucose uptake with *GLUT* gene expression levels within seven patients. Not surprisingly, we found that *GLUT1* expression is highly correlated (Spearman correlation $\rho = 0.5$) with glucose uptake in epithelial cells, and *GLUT3* expression correlated with (Spearman correlation $\rho = 0.39$) glucose uptake in myeloid cells (Fig. 5g). Somewhat surprisingly, we found that *GLUT3* expression is also highly correlated (Spearman correlation $\rho = 0.46$) with glucose uptake in epithelial cells, which implies that it could potentially mediate glucose metabolism in epithelial cells. Overall, METAFlux analysis revealed metabolic heterogeneity among various cell types in LUAD patient cohorts and nominated metabolic genes or reactions that could be further examined in functional studies.

### METAFlux characterized metabolic competition between tumor and CAR-NK cells

To further examine METAFlux's applicability in studying cancer immunotherapy, we analyzed scRNA-seq data generated from an in vivo experiment of engineered CAR-NK cells with Raji, a non-curative CD19+ lymphoma cell-line in immunocompromised NSG mice[76,77]. In the Raji/NK experiment, NK cells were purified, transduced, and injected into mice for in vivo experiments, with survival and tumor cell quantification conducted. More detailed protocols are available in our previous work[76]. Assessed were three CAR-NK cell products: CAR19/IL15 NK cells armed with IL-15, CAR19 NK cells lacking IL-15, and non-transduced NK (NT-NK) cells at four time points: day 7, day 14, day 21, and day 28.

We ran METAFlux in a community-based setting (Methods) using cell type proportions estimated from the scRNA-seq data. The METAFlux prediction at Day 7 indicated that the three NK cell products interacted with tumor cells via competition, cooperation, or joint release of a variety of metabolites (Fig. 6a). Among these, CAR19/IL15 NK cells had the highest oxygen consumption and proton efflux levels, followed by CAR19 and NT NK cells (Fig. 6b). To validate this result, we performed Seahorse assays using Agilent Seahorse XFe96 analyzer, which measured OCR, a proxy of oxygen consumption flux and ECAR, a proxy of proton efflux[78] two hours after coculturing NK and Raji cells ($N = 3$). More details regarding the Seahorse assay protocol can be found in our previous study[76]. We found a highly consistent trend between METAFlux prediction and Seahorse results (Fig. 6b, c). For basal respiration, the Hedges'g between CAR19/IL15 and CAR19 is 1.63 with 95% CI [−0.47, 2.72]. For glycolysis, the Hedges'g between CAR19/IL15 and CAR19 is 2.30 with 95% CI [−0.04, 4.62]. For both basal respiration and glycolysis, the Hedges'g estimates are considered large sizes. This suggests that the trend is robust and statistical significance was not achieved due to the limited sample size. We then investigated metabolism of different products using per cell competition score (PCCS), defined as the ratio of the per cell nutrient uptake flux in the tumor cells over that in the NK cells. We found CAR19IL15 NK cells experienced the least oxygen competition, while the NT NK cells experienced the highest (Supplementary Fig. 5a). Interestingly, CAR19/IL15 cells also had the highest glucose uptake level among the three products at day 7, indicating that CAR19/IL15 NK cells were the most metabolically active (Supplementary Fig. 5b).

We further examined the metabolic changes over time under the CAR19/IL15 product. We found that the PCCS for oxygen and glucose decreased from day 7 to day 14 but ramped up after day 14 and peaked on day 28 (Fig. 6d). Similar trends were observed in the PCCS of several amino acids (Supplementary Fig. 5c–f). These results suggested that

tumor cells eventually outcompeted NK cells in terms of nutrient uptake and that NK cells became less metabolically fit over time, which is consistent with the observed phenotype[76].

We also observed that tumor cells increased lactate production in the TME over time, which peaked on day 21 and day 28 (Fig. 6e and Supplementary Fig. 5g). Similarly, NK cells had elevated lactate release over time (Supplementary Fig. 5h). As a result, the TME became increasingly acidic, which might have suppressed the cellular function and cytotoxic activity of NK cells, leading to tumor recurrence[79]. Overall, our findings indicated that metabolic competition in TME likely contributed to tumor resistance and relapse and that METAFlux provided mechanistic insights on specific metabolites and cell-types that cannot be readily measured via experimental means.

## Discussion

Despite advances in new technologies, performing comprehensive metabolic profiling is still limited by coverage, robustness, and cost. Although computational methods have been developed to characterize metabolic traits from RNA expression data, the reliability of these methods is unclear, particularly when applying to single-cell settings. In this work, our method METAFlux appears as a viable solution that enables a broad spectrum of metabolic flux characterization from bulk, flow-sorted and single-cell RNA expression data, in a medium and TME community-aware manner, which has led to deeper insights on metabolic subtypes of cancer patients, metabolic heterogeneity in TME, and mechanisms of tumor resistance in immunotherapeutic settings. Of particular interest, METAFlux enables characterization of metabolism architypes and heterogeneity at cell-type resolution in various cancer datasets under a community-based model, which has not been possible before using bulk data and existing modeling methods.

We expect that METAFlux will enhance hypothesis generation and validation from perspective of metabolomics and facilitate translation of new therapies into clinical trials. The recent pioneering discovery of JHU083, a glutamine-antagonizing drug that can suppress tumor growth and promote anti-tumor T cell function, showcases the great importance of disentangling the metabolism of cancer cells from that of immune cells[16]. However, an avenue toward developing metabolism-based drugs is very challenging, partly because of the shared metabolic pathways among cancer cells, immune cells, and normal cells. Targeting specific metabolic pathways can often inhibit tumor growth but also derail immune cell function. In this regard, METAFlux holds great promise in exploring clinically differential metabolic targets, because it can account for interaction dynamics (architypes and PCCS, etc.) among multiple cell types residing in TME, aiding the process of discovering differential or coordinated metabolic responses.

Although the strengths of METAFlux have been successfully demonstrated, some limitations remain. First, the predicted glucose uptake does not correlate as consistently as other metabolites in NCI-60 panel. This discrepancy potentially results from the limitations of the cell-line models which do not accurately encapsulate the multifaceted nature of glucose metabolism. Alternatively, it suggests that the methodology implemented by METAFlux need improvement to fully capture the intricacies of metabolic processes under certain scenarios. Second, METAFlux does not distinguish different levels of nutrient concentrations and only utilized binary values (presence vs. absence) in specifying medium constraints. Although this feature can be advantageous when detailed experimental parameters are not available, there can be situations where concentrations or fluxes of certain nutrients vary in wide ranges and can significantly affect accuracy of the results. Thus, improving the approach so that it can further leverage the granular medium conditions may be required to further improve the prediction accuracy.

Third, current objective function of METAFlux maximizes the flux of generic biomass reaction (*biomass_human* from Human1), which

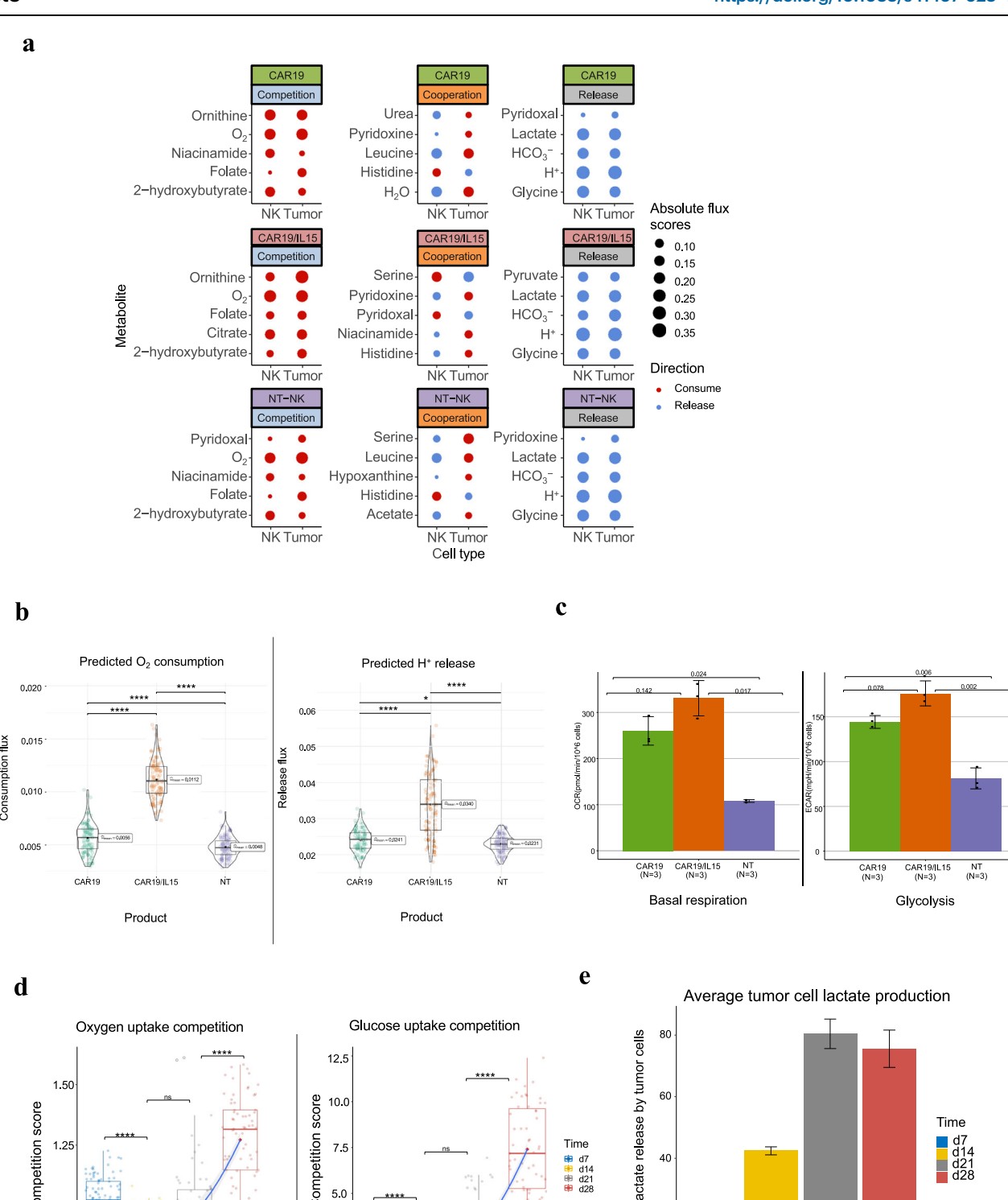

encompasses various cellular components for cellular proliferation including, for example, proteins, lipids, small metabolites, etc. This formulation has been proven to be a fairly accurate characterization of generic human cell composition and demand[53]. However, it may be more accurate to employ a cancer-specific biomass formulation, assuming tumor cells are the ones driving biomass growth in a TME.

Fourth, despite the high resolution provided by single-cell data, the current version of METAFlux calculates metabolic fluxes at cell cluster level, due primarily to considerations on data sparseness and noisiness of scRNA-seq data at individual cell level. Direct estimation of cell-wise MRAS from zero-inflated data can result in many zeros, which challenges downstream modeling. Averaging gene expression at

**Fig. 6 | CAR-NK single-cell RNA-seq METAFlux analysis. a** Top five nutrients profile of NK and tumor cells for each product under each metabolism architype. The top five nutrients were selected by the most significant *P*-values from ANOVA analysis of NK cell fluxes. Dot size represents the mean absolute cubic root normalized flux scores, and color represents the direction of flux. **b** Violin plots of predicted oxygen consumption flux and H⁺ release flux for the day 7 product of CAR19/IL15, CAR19, NT-NK. Each group includes $n = 100$ bootstrap samples. The box within represents the interquartile range (IQR), the line within shows the median and the whiskers extend up to 1.5IQR. Min and max values correspond to the smallest and largest mean predicted flux after removing data points that fall more than three times the IQR beyond the quartiles (Q1 and Q3). Statistical test: two-sided Games-Howell test. *P*-values are Holm corrected. *P*-values are denoted as follows: ns ($p > 0.05$), *($p \leq 0.05$), **($p \leq 0.01$), ***($p \leq 0.001$), ****($p \leq 0.0001$). Oxygen consumption: Adjusted *P*-value = $1.0 \times 10^{-13}$ for CAR19 vs CAR19/IL15. Adjusted *P*-value = $1.0 \times 10^{-5}$ for CAR19 vs NT-NK. Adjusted *P*-value = 0 for CAR19/IL15 vs NT-NK. H⁺ release: Adjusted *P*-value = 0 for CAR19 vs CAR19/IL15. Adjusted *P*-value = 0.03 for CAR19 vs NT-NK. Adjusted *P*-value = $6.7 \times 10^{-14}$ for CAR19/IL15 vs NT-NK. **c** Bar graphs of mean basal respiration and glycolysis obtained respectively from the Seahorse assays. Error bar: mean ± sd. Pairwise test: two-sided Games-Howell test. **d** Per cell competition score (PCCS) between cancer and NK cells in TME from day 7 to day 28 for oxygen and glucose uptake, respectively for CAR19/IL15. Test: *T*-test. PCCS is defined as the ratio of the per cell nutrient uptake flux in the tumor cells over that in the NK cells. Each group includes $n = 100$ bootstrap samples. The box represents the interquartile range (IQR), the line within shows the median, and the whiskers extend up to 1.5IQR. Min and max values correspond to the smallest and largest mean predicted flux after removing data points that fall more than three times the IQR beyond the quartiles (Q1 and Q3). The grey band represents the 95% confidence interval for the loess (locally weighted scatterplot smoothing) curve. *P*-values were calculated using the two-sided Wilcoxon Rank Sum test and corrected by FDR. *P*-values are denoted as follows: ns ($p > 0.05$), *($p \leq 0.05$), **($p \leq 0.01$), ***($p \leq 0.001$), ****($p \leq 0.0001$). Oxygen consumption competition: Adjusted *P*-value = $2.3 \times 10^{-6}$ for time point 7 vs time point 14. Adjusted *P*-value = 0.06 for time point 14 vs time point 21. Adjusted *P*-value = $1.6 \times 10^{-9}$ for time point 21 vs time point 28. Glucose uptake competition: Adjusted *P*-value = $1.5 \times 10^{-7}$ for time point 7 vs time point 14. Adjusted *P*-value = 0.05 for time point 14 vs time point 21. Adjusted *P*-value = $6.3 \times 10^{-10}$ for time point 21 vs time point 28. **e** A bar plot of tumor cells' contribution to total lactate release in TME over time. The *Y*-axis denotes the percentage of lactate released by the tumor cells. Test: *T*-Test. Error bar: mean ± sd. Each time point includes $n = 100$ bootstrap samples.

cluster level can substantially alleviate such negative impacts, although it is less effective for smaller clusters. For datasets with high dropout rates, it may be beneficial to apply gene expression imputation on cell clusters of a minimal size before applying METAFlux. For example, the Compass metabolic model, recently developed by Wanger et al., used KNN smoothing to mitigate sparsity and stochasticity[80]. Such modeling efforts will be further required to fully harness the power of single-cell data. A more in-depth comparison between METAFlux and Compass is presented in the Supplementary Note.

In summary, we have demonstrated that METAFlux can achieve accurate, broad-spectrum characterization of metabolic fluxes from bulk and single-cell transcriptomic data. The computational time of METAFlux varies for bulk and single-cell settings. On a bulk dataset with 500 samples, it typically takes 18 minutes and a maximum of 4.4 GB memory to complete the whole pipeline on a desktop computer with 3.3 GHz Dual-Core Intel Core i7 processor. On a single-cell dataset with a total of 10,000 cells (bootstrap number = 50 and number of cell type = 2), the elapsed time was 17 minutes, and the maximum memory usage was 12 GB. If widely used, METAFlux can enable systemic, accurate quantification of cancer metabolism in numerous studies and generate insights on metabolic heterogeneity and mechanisms across patients and cell types, potentially driving discoveries for cancer therapy.

## Methods

### Underlying genome-scale metabolic model (GEM)
Human1, available publicly, contains 13,082 reactions and 8378 metabolites[53]. Reactions are across nine compartments (extracellular, peroxisome, mitochondria, cytosol, lysosome, endoplasmic reticulum, Golgi apparatus, nucleus, inner mitochondria).

### Inference of metabolic reaction activity score (MRAS) from transcriptomic data
We use GPR rules, which decode Boolean logic relationship between genes in a reaction[81], to map the relationship between gene products and then summarize a total of 3625 metabolic related gene expressions into Metabolic Reaction Activity Scores (MRAS) given the predefined relationships. We do not consider reaction kinetic constants and binding affinity of proteins, since it is difficult to robustly estimate all these parameters in genome-scale modeling[45,82]. Our approach is adopted from what has been proposed earlier to infer activity of metabolic reaction from gene expression data[43,83].

In GPR, the AND operator joins the genes encoding for different subunits of the same enzyme, and the OR operator joins the genes

encoding for isoenzymes[84]. For a reaction catalyzed by an enzyme complex, all the subunits need to be expressed to catalyze a reaction, and the lowest expressed unit will be the rate-limiting step for this complex. Therefore, the metabolic activity of such an enzyme complex will be the lowest expression value among all genes associated with this enzyme complex. For a reaction catalyzed by isoenzymes, all the isozymes contribute additively to this reaction[83]. Thus, metabolic activity will be the sum of all expressions of isoenzyme genes. Some genes are involved in multiple reactions (e.g., promiscuous enzyme), and we hypothesize that there may be enzyme resource competition may exist between reactions. We adjust for the enzyme promiscuity by dividing the expression value of a gene by the number of reactions the gene has participated in. A similar approach has been shown in[85]. The steps of deriving $MRAS_{ij}$, metabolic reaction activity for the *j*th reaction for the *i*th sample are the following:

Let $w_k$ be the number of reactions enzyme $k$ participates in for the *j*th reaction and $Enzyme_{ik}$ be the gene expression of $Enzyme_k$ in sample *i*, then

$$OR\ logic : MRAS_{ij} = \sum_{k=1}^{n} \frac{Enzyme_{ik}}{w_k} \tag{1}$$

$$AND\ logic : MRAS_{ij} = Min \left[ \frac{Enzyme_{i1}}{w_1}, \frac{Enzyme_{i2}}{w_2}, \frac{Enzyme_{i3}}{w_3} \dots \right] \tag{2}$$

### Defining reaction flux constraints using gene expression derived MRAS
To connect transcriptome and fluxes, a possible solution is to use the MRAS calculated before to define the flux constraints. We use an approach similar to E-flux[45], where the expression levels of the genes associated with a metabolic reaction serve as the maximum possible flux that the reaction can carry. The rationale is that, although enzyme activities do not have a high correlation with RNA levels, given a specific level of translational efficiency and assuming there is a limited accumulation of enzymes over a certain time window[45], RNA expression levels can be used as the maximum amount of protein products available, which can then serve as the maximum reaction fluxes.

We set normalized MRAS as the flux upper bound to their corresponding metabolic reactions. The lower bound of reaction flux is set zero, if the reaction is non-reversible and $(-normalized\ MRAS)$, if the reaction is reversible. The flux is loosely constrained when MRAS is high, so there is more bandwidth of reaction flux. On the other hand,

the flux is strictly constrained when MRAS is low, so the bandwidth of flux is much narrower. Our model consists of 13,082 metabolic reactions in total, and 8033 reactions are associated with enzymes (8033 R1 reactions). The rest of the reactions are not associated with any enzymes (5049 R2 reactions) and thus are not constrained by gene expression, so we set their corresponding upper bounds to one.

The constraints of flux are as follows:

$$(lb_{ij}, ub_{ij}) = \begin{cases} lb_{ij} = 0, & ub_{ij} = \frac{MRAS_{ij}}{\max(MRAS_i)} & \text{if } rev_j = 0 \text{ and } j \in R1 \quad (3) \\ lb_{ij} = -\frac{MRAS_{ij}}{\max(MRAS_i)}, & ub_{ij} = \frac{MRAS_{ij}}{\max(MRAS_i)} & \text{if } rev_j = 1 \text{ and } j \in R1 \quad (4) \\ lb_{ij} = 0, & ub_{ij} = 1 & \text{if } rev_j = 0 \text{ and } j \in R2 \quad (5) \\ lb_{ij} = -1, & ub_{ij} = 1 & \text{if } rev_j = 1 \text{ and } j \in R2 \quad (6) \end{cases}$$

where $lb_{ij}$ and $ub_{ij}$ are the input flux bounds for the $jth$ reaction in the $ith$ sample, and $rev$ stands for the reversibility of a reaction. If $rev = 0$, the reaction is non-reversible; if $rev = 1$, the reaction is reversible.

## Optimization framework

**Stoichiometric representation of metabolic reactions.** A chemical reaction is a basic unit in metabolic pathways, and stoichiometry can represent the quantitative relationship between products and reactants in a reaction. The stoichiometric coefficients of the reactions populate the stoichiometric matrix **S**. Here, Stoichiometric matrix is an **M** by **N** sparse matrix, where **M** equals 8378, the number of metabolites in different compartments, and **N** equals 13,082, the total number of metabolic reactions. The negative coefficient in **S** matrix refers to the moles of metabolites consumed in a particular reaction. The positive coefficient means how many moles of the metabolites are produced in a specific reaction. At the same time, zero implies that this metabolite does not participate in a specific reaction.

**Defining nutrient availability profile.** We use *the cell line* culture medium (Supplementary Data File S1), containing 44 metabolites as the growth medium in cell line models[53]. These 44 metabolites include major components from Hams F-12 medium (amino acids and vitamins) and other essential nutrients and ions from serum supplements (Supplementary Methods). At this point, we only offered a binary list of significant metabolites present in the environment. The reason we need to define the metabolite availability is to guide the optimization search in a biologically relevant sub-space. The uptake or secretion rates of these 44 metabolites are not limited. For the remaining metabolites in the model, we do not allow cells to uptake them from the medium but instead, allow cells to secrete them into the medium. For tissue samples from patients, it is necessary to define a more physiologically relevant environment, as the traditional synthetic medium does not mimic *human blood*. Cantor et al. developed a human plasma-like medium [HPLM] to better capture the composition of *human blood*, and we derived a list of 64 metabolites in *human blood* based on their profiling (Supplementary Data File S1)[17]. For more general use purposes, users are not restricted to the pre-defined metabolite list. We have provided the option to modify the input metabolite list if users have knowledge of their background metabolite profiles.

**Solving flux balance analysis using quadratic programming.** Traditional FBA by linear programming (LP) gives a unique optimal objective value. However, the solution to FBA from LP is most likely degenerate, meaning the solutions are not unique, and different solvers will likely return different vectors. Usually, the flux variability analysis (FVA) is used afterward to calculate the range of fluxes that achieves the optimal objective[86]. Another approach, pFBA or Parsimonious enzyme usage FBA, was proposed earlier[34]. pFBA assumes there is a selection for an organism to minimize the total amount of necessary enzymes to achieve optimal growth. pFBA first computes the optimal growth rate and then minimizes the sum of reaction fluxes under the optimal solution. Here, we reformulate this idea into convex quadratic programming (QP) to overcome degeneracy.

We define single-sample unsolved metabolic fluxes by vector **v** with a length of 13,082. The dot product of matrix **S** and a vector of unknown fluxes **v** approximation to 0 represents the steady-state assumption, where the metabolite concentrations are essentially held constant $\frac{dx}{dt} \approx 0$, and $x$ stands for the concentration of all the metabolites.

The framework is implemented using OSQP solver[87] and formulated as the following optimization problem:

$$\min \frac{1}{2} v^T v - a C^T v \qquad (7)$$
$$s.t. \, Sv \approx 0$$

$$lb_j \le v_j \le ub_j \qquad (8)$$

$$lb_j = 0 \text{ if } j \in E \text{ but } j \notin GE \qquad (9)$$

where $C$ is a vector of zeros with a one at the position of our designated biomass reaction, and constraint (9) are the growth medium constraints. $a$ is set to 10,000, the same order of magnitude with respect to fluxes used in Fit methods[54]. $j$ stands for the $jth$ reaction. We define a growth medium $G$ and all exchange reactions $E$ and $GE$ as exchange reactions relevant to $G$. Usually, fluxes $v$ are in units like mmol/gDW/hr, but our predicted fluxes are inferred from gene expression. Thus, the results are relative flux scores. To retrieve the uptake or release flux of a specific metabolite, we query the exchange flux using the reaction ID. A positive flux indicates secretion, whereas a negative flux represents the uptake of the metabolite.

## Single-cell pipeline: community-based flux estimation in single-cell data

Given a scRNA-seq dataset, we assign a group (cluster or cell type) label to each cell. We first perform stratified bootstrapping, which means we sample with replacement with respect to each group.

Step 1: Bootstrap sample generation. Let group be $g = 1, \dots n$, and $b_{ig}$ the $ith$ bootstrap sample for the $gth$ group. For each bootstrap iteration $i$, we combine $b_{i1}, b_{i2} \dots$ for all the groups to form resampled data $B_i$. Each generated bootstrap data will be the same size and have the same group proportion as the original data.

Step 2: MRAS calculation on bootstrap samples. For each bootstrap sample $B_i$, we calculate the mean gene expression vector for each group. Next, we calculate the MRAS for each mean gene expression vector. This step is same with bulk pipeline. MRAS will be later used to derive $lb_{ij}$ and $ub_{ij}$, where $ij$ stands for the $jth$ reaction in cell group $i$.

Step 3: Define group fraction parameter $P$. $P$ is a fraction matrix defined as:

$$P = \begin{bmatrix} p_1 & 0 & 0 & \cdots & 0 \\ 0 & p_2 & 0 & \cdots & 0 \\ 0 & 0 & p_3 & \cdots & 0 \\ \vdots & \vdots & \vdots & \ddots & \vdots \\ 0 & 0 & 0 & \cdots & 1 \end{bmatrix} = diag(p_1, p_2 \dots, 1) \qquad (10)$$

where $p_i$ stands for the percentage of cell group $i$, and that $\sum_{i=1}^n p_i = 1$.

We require group fractions as our input. Group fractions indicate the proportions of groups of interest with respect to the whole sample. Ideally, these proportions should be retrieved from experiments or calculated from matched bulk data using CIBERSORTx[88]. However, most datasets do not have such information. As a result, directly observed cluster (group) fractions in single-cell data could be used, but further studies are warranted to evaluate the findings since those

proportions may deviate from the truth due to bias introduced during tissue dissociation[89,90].

Step 4: Merging metabolic networks. Let $m_{ij}^E$ be the metabolites associated with exchange reactions in group $i$. To merge multiple metabolic networks, we need to create a "TME metabolite reservoir" for different cell groups to interact. We define $rm_{ij}$ as the reservoir metabolite $j$ in group $i$

$$C_i : m_{ij}^E \leftrightarrow rm_{ij} \tag{11}$$

which allows different partitions of TME to share the same resources, and

$$E(SR) : rm_{ij} \leftrightarrow \varnothing \tag{12}$$

which ensures the model is an open system that allows reservoir metabolites to be exchanged with the external environment. Each $S_i$ represents the cell group specific stoichiometric matrix. The final size of the merged stoichiometric matrix is $(N \times 8378 + 1648) \times (13,082 \times N + 1648)$, where $N$ is the number for groups we defined. A specific construct of the merged stoichiometric matrix for three groups is shown below:

|  | Group 1 reactions | Group 2 reactions | Group 3 reactions | Shared metabolic reactions |
|---|---|---|---|---|
| Shared metabolites | $C_1$ | $C_2$ | $C_3$ | $E(SR)$ |
| Group 1 Metabolites | $S_1$ | $O$ | $O$ | $O$ |
| Group 2 Metabolites | $O$ | $S_2$ | $O$ | $O$ |
| Group 3 Metabolites | $O$ | $O$ | $S_3$ | $O$ |

Our model aims to maximize the entire community's biomass while minimizing the sum square of overall fluxes.

Step 5: Flux calculation. This step is implemented using OSQP solver[87]. For each bootstrap sample mean expression vector, it is formulated as follows:

$$\min \frac{1}{2} v^T P v - a C^T P v \\ s.t. \ SPv \approx 0 \tag{13}$$

$$lb_{ij} \leq v_{ij} \leq ub_{ij}, if \ j \notin E(SR) \tag{14}$$

$$lb^e \leq v_j^e \leq ub^e, if \ j \in E(SR) \ then \ ub^e = 1 \tag{15}$$

$$lb^e = 0 \ if \ j \in E(SR) but \ j \notin GE(SR), otherwise \ lb^e = -1 \tag{16}$$

$C$ is a vector of zeros with ones at each cell group's designated biomass reaction position. $a$ is set to $10,000$. $j$ stands for $jth$ reaction in $E(SR)$ or shared exchange reactions. $ij$ stands for the $jth$ reaction in cell group $i$. Constraint (15) and (16) are the constraints imposed specifically on shared exchange reactions or $E(SR)$. Constraint (16) represents the growth medium constraint, and we define a growth medium $G$, and $GE(SR)$ as shared exchange reactions relevant to $G$. For bulk-sorted data, we do not perform any bootstrap. We will directly calculate MRAS and proceed to step 3 and follow the rest of the pipeline.

## Identification of metabolic interaction modes in single cell pipeline

TME is a highly complex mixture, and TME components either form metabolic antagonism or symbiosis when uptaking nutrients[91]. When one or more cell types benefit from the metabolites produced by other cell types, we define the interaction as a metabolic *Cooperation* program. When all cell types compete for limited resources in TME, we define it as a *competition* metabolic program. When all cell types secrete certain nutrients, we define it as a *Release* program.

## Simulation of metabolic fluxes using random medium profiles

To estimate the medium effect on model performance, we compare the results obtained from our biologically meaningful medium with same-sized random mediums. The assumed medium contains 44 metabolites, so we randomly select 44 metabolites from the total 1648 exchange metabolites. We then allow our model to uptake or secrete those 44 metabolites without rate restriction, while allowing the rest of 1604 metabolites to only secrete with no rate restriction. We repeat this process $N = 500$ times. For each simulation $i$, we obtain the overall Spearman correlation $\rho_i$ and directionality accuracy $acc_i$. To calculate the p-value, we count the number of measurements greater than our original biological meaningful statistics $\rho_o$ and $acc_o$ and divide this number by 500.

$$pvalue_\rho = \frac{\sum_{i=1}^{i=N} I(\rho_i \geq \rho_o)}{N} \tag{17}$$

$$pvalue_{acc} = \frac{\sum_{i=1}^{i=N} I(acc_i \geq acc_o)}{N} \tag{18}$$

Where $I(.)$ Is an indicator function, which equals to one if the condition in parenthesis is true, and zero, otherwise.

## Statistics and reproducibility

The sample size for our study was not predetermined through a statistical method. The sample sizes were primarily decided based on the availability of public data. The number of bootstraps performed was mainly driven by the computational time frame and resources. Detailed protocols for analyzing flux data and gene expression data are described in the Supplementary Methods.

## Reporting summary

Further information on research design is available in the Nature Portfolio Reporting Summary linked to this article.

## Data availability

All datasets used in our study are publicly available. Human-GEM model was accessed from (https://github.com/SysBioChalmers/Human-GEM). We retrieved the medium composition and experimental flux profiling data for 11 NCI-60 cell lines under the original manuscript[53]. ecGEM flux prediction for 11 NCI-60 cell lines was be obtained at https://zenodo.org/record/3583004#.YhQJdZPMJqs. NCI-60 cell lines TPM RNA-seq data was obtained from https://depmap.org/portal/download/. TNBC FDG-PET data and matched expression data can be found at NCBI Gene Expression Omnibus (GSE135565), and pancreatic cancer FDG-PET data and matched expression can be downloaded at NCBI Gene Expression Omnibus (GSE107754). The TCGA pan-cancer RNA-seq TPM data was downloaded from UCSC Xena data hubs (https://xenabrowser.net/). The proliferation score data can be found in the original publication[64]. Patient Lung cancer bulk-sorted RNA-seq data can be downloaded from the NCBI Gene Expression Omnibus (GSE111907). Single-cell Metastatic LUAD data was retrieved from NCBI Gene Expression Omnibus (GSE131907). CAR-NK single-cell RNA-seq data is available through NCBI Gene Expression Omnibus (GSE190976).

## Code availability

Source code is publicly available at https://github.com/KChen-lab/METAFlux or https://zenodo.org/badge/latestdoi/515741372.

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

## Acknowledgements

We thank Dr. Brooks Phillip Leitner for sharing processed TCGA LUAD SUV data. This project has been made possible in part by grant RP180248 to K.C. from Cancer Prevention & Research Institute of Texas, grant U01CA247760 to K.C. and the Cancer Center Support Grant P30

CA016672 to P.P. from National Cancer Institute, and Human Cell Atlas Seed Network Grant CZF2019-02425, CZF2019-002432, and CZF 2021-239847 to KC from the Chan Zuckerberg Initiative DAF, an advised fund of Silicon Valley Community Foundation. This project was also partially supported by the MD Anderson Moonshot programs.

## Author contributions

Y.H. and K.C. conceptualized and designed the study. Y.H. and K.T. implemented the METAFlux tool. K.C. supervised the project. Y.H., V.M., M.D. (Merve Dede), K.T., and K.C. performed data analysis and interpretation. May Daher, L.L., and K.R. specifically contributed to CAR-NK data generation, curation, and analysis. Y.H., V.M., Merve Dede, and K.C. drafted the manuscript, incorporating input from all authors. All authors reviewed, edited, and approved the final manuscript.

## Competing interests

The authors declare no competing interests.
