## [Peer Review File · Nature Communications]

REVIEWER COMMENTS

Reviewer #1 (Remarks to the Author): expertise in lung cancer metabolism

Here, Huang et al invented a novel computational framework, METAFflux (METAbolic Flux balance analysis) to infer metabolic fluxes in cancers from transcriptomic data. The authors included a workflow for bulk and single cell RNA seq data, and addressed the interaction of different cell types in the tumor microenvironment. Moreover, nutrient availability, as an important constraint for metabolic fluxes, was considered, however in a binary fashion (nutrient present or absent in plasma or in cell culture medium). The study is timely, metabolism prediction tools from expression data could be highly useful to the field, if their limitations would be taken into account. In the present study, the prediction of the relative contribution of different cell types in the TME to the uptake/release of different metabolites in lung adenocarcinoma is of interest and the time-dependent changes of metabolism in the CAR-NK cell treatment (immunotherapy) model provide novel insights. Although the results of the study certainly demonstrate the potential of this platform to analyze tumor metabolism and metabolic interactions between cancer cells and the TME, the model "validation" is largely based on application of the model to various datasets. As a first approach the authors aimed to validate their model using the published dataset from the study on Human1, the genome-scale metabolic model which served as the basis for the present paper. The dataset comprised release or uptake data of metabolites of 11 cancer cell lines in vitro (ref 52). Uptake/release rates correlated with the predicted uptake values for the majority of metabolites analyzed, however, the model appears to show a weakness in predicting glucose uptake, which is a key feature in cancer cells and e.g. T cells. Moreover, the authors performed Seahorse measurements using Raji lymphoma cells co-cultured with NK cells, yet, a detailed methods description for the Seahorse assay is missing. As a weakness of the study, central metabolic pathways like glycolysis-branching pathways (e.g. the pentose phosphate pathway), amino acid, nucleotide or lipid synthesis pathways, or the TCA cycle were apparently not validated. This would require more elaborative assays, like stable isotopic tracing, still some effort could have been made in validating predictions for these pathways as well. Other possible approaches include essentiality prediction and experimental verification. As another limitation of the study, the real extracellular tumor microenvironment was not modeled. As the authors discuss, this is at present a difficult task, due to a lack of extracellular fluid metabolomics data for most tumors.

#Major points

#1. Fig. 2a and b: Correlations of predicted uptake/release rates for 26 metabolites in 11 different cell lines from the NCI-60 panel is shown, with an overall positive Spearman correlation coefficient. However, glucose and lactate, highly important metabolites in cancer, show no or even a negative correlations between predicted and measured uptake/release rates, although the experiments are performed in defined conditions in vitro. The model might be even poorer in predicting glucose and lactate in vivo fluxes in complex environments. Still, the remainder of the paper highly focusses on glucose uptake. How did the authors validate that glucose uptake is sufficiently well predicted by the model, except for the comparison of key results with published data (e.g. higher Glucose uptake in lung squamous cell carcinoma compared to lung adenocarcinoma, and high uptake by myeloid cells, Fig 4)? Fig. 1c suggests that the model seemingly well predicts the direction of flux of most metabolites, however, when comparing different cells, treatment, or tissues also the extent of flux, not only the direction (uptake or release), are important

#2. The approach by Huang et al to include physiological constraints rather than modelling metabolism based on fluxes derived from in vitro data alone is a potential advantage. Still, as the authors state in the Discussion, the model is based on the absence or presence of metabolites (binary variable) instead of including available metabolite concentrations, which is a limitation. The authors address this issue in the discussion. To address this issue, the paper could be improved by assessing METAFflux results in tumors showing a very low perfusion/nutrient abundance, e.g. pancreatic cancer, and to test the accuracy of the model prediction (as shown in Fig. 2b) e.g. using tumor cells or immune cells cultured in plasma-like medium or starvation medium. Does the model still provide meaningful results in such situations? The authors should explicitly mention this limitation also in the online version of METAFflux and in which situations this may lead to incorrect results. Moreover it should be made clear, for which cell culture medium the framework is designed, since they vary widely (e.g. DMEM medium does not contain certain amino acids, like

aspartate or asparagine).

#3. A detailed description of this profile of "consumable" metabolites is missing. According to the methods (section 4.4.2.), the list of consumed or secreted metabolites contains 44 metabolites present in cell culture growth medium, which are not limited. The model does not allow for uptake of the remaining metabolites, the ones that are limited, as is stated in methods 4.2.2. However, usually it is the rapid uptake of metabolites rendering them limiting in vitro, thus it appears unclear, why the model is designed in a manner not allowing the cells to uptake these "remaining" metabolites from the medium (see line 445). Glucose is a limiting factor according to Ref 52, so thus the model by Huang et al exclude glucose uptake? How is it then computed in the analyses in the rest of the paper?

#4. Hypoxia is another important modulator of cell metabolism and is frequently present in tumors. Single cell RNA seq may potentially allow to include hypoxia in the model, however, it is not clear, whether this has been done.

#5. Some results are interpreted by the authors in a manner suggesting that the actual glucose uptake rates are known. E.g. line 221, results: "However, a significant difference [of glucose uptake] emerged in immune cells ($P=0.045$, 95% CI of effect size $[-1.8626, -0.1579]$, two-sample t-test), indicating that immune cells may not be deprived of glucose in the TMEs of the LUSCs". Gene expression certainly may predict the metabolic phenotype, as has been shown in several studies, however, whether this applies to glucose-limited environments is less well understood. This should be pointed out.

#6. Certain metabolites, which are part of competing metabolic pathways but also can be released (e.g. citrate) are included in the cell-cell interaction model. How did the authors estimate release of such central metabolic pathway intermediates based on gene expression? A description of the underlying assumptions should be given and an experimental validation of citrate (or succinate) exchange with the extracellular space (which is of high interest in the field) should be ideally done.

#7. The differences to and advantages over the approach used by Alghamdi et al DOI: 10.1101/gr.271205.120, a recently published network model to estimate cell-wise metabolic flux using single-cell RNA-seq data, should be discussed.

#8. Although metabolic pathways are complex and interconnected, the model may be flawed by including genes into a pathway that in fact mediate the opposite pathway. E.g. in Supplementary Table S2 ("Glycolytic pathway genes"), FBP1, FBP2 are included, which are not glycolytic but exclusively gluconeogenic (=reverse pathway). Also, G6PC, G6PC2 and G6PC3 do not mediate glycolysis, but gluconeogenesis or glycogenolysis. The authors should re-examine their pathway genesets manually for misannotations.

#9. Fig. 6c: are the differences significant?

Minor points:

1. Line 238: TME cells cannot "produce" phenylalanine, strictly speaking, since phenylalanine is an essential amino acid. They can just release it from their stores (e.g. after proteolysis).

2. Suppl Table S1: Ions are not nutrients, thus the caption should be changed. Reformatting the table (Superscripts and Subscripts) is needed. Also oxygen is not a nutrient (Main text line 236).

3. No method description for the Seahorse assay and for the Raji/NK cell co-culture experiment is given. How were the NK cells obtained?

4. Fig. 2b: How is biomass defined and why was this parameter included in metabolite uptake/release assays?

5. Introduction, line 57 "13C metabolic flux analysis (13C-MFA) and Seahorse Extracellular Flux (XF) analyzer, have become widely used in metabolic research. 13C MFA is the current gold standard in measuring intracellular fluxes of central carbon metabolism and to assess cells'

extracellular bioenergetic state. It can measure OCR (Oxygen consumption rate: an indicator of mitochondrial respiration) and ECAR (extracellular acidification rate": The last sentence refers to the XF Analyzer, not to 13C-MFA. 13C-MFA can assess metabolic fluxes based on label enrichment in key metabolites following incubation with stable isotopic tracers (such as 13C-labeled precursors).

6. Introduction: To complete the summary on existing approaches to assess metabolism in human or model tumors, *in vivo* stable isotopic labelling should be mentioned, not just PET (which PET?), see e.g. Faubert, Berardinis RD et al, doi: 10.1038/s41596-021-00605-2.

7. Methods, 4.4.2. (line 442), Defining nutrient availability profile for cell culture and patient samples: It is not clear, which 44 metabolites were selected to be defined as "non-limiting" and why. In order to make the constraint model more clear, the authors should not just state "We use the cell line culture medium, containing 44 metabolites as the growth medium in cell line models (Ref 52)", but provide a list of these 44 metabolites. Ref 52 reports on the use of Ham's medium for their model, however which Ham medium was used (Ham F-10 or F-12) is not specified), also if serum was added to the medium.

8. Ref 6 is an abstract. Given the large body of literature on the role of metabolic perturbations on cancer aggressiveness the authors may find different literature to fit to their introduction.

9. Fig. 3d and S2f: How is overall survival defined? This should be stated in "methods". Patients at risk or total number of patients

10. Section 2.4. It should be "18F-FDG (fluorodesoxyglucose) PET, not "FGD-PET".

Reviewer #2 (Remarks to the Author): expertise in cancer metabolomics flux analysis

This manuscript "Characterizing cancer metabolism from bulk and single-cell RNA-1 seq data using METAFlex" reported a new computational framework METAFlex (METAbolic Flux balance analysis), which can infer metabolic fluxes from bulk or single-cell transcriptomic data. The authors clearly summarized the current approaches for analyzing the metabolic data and their limitations. The introduction is well-written, and the authors pointed out the gaps in the current approaches. However, it is unclear how the current METAFlex is improved compared to the previous ones, such as ecGEMs, as figure 2c shows there are largely similar. Metabolically, this study highly focuses on glucose uptake (Fig2-5). Since glucose uptake can be directly measured, it is unclear if and how the current METAFlex can provide accurate information about the other metabolism. Together, although this is an interesting study, which is trying to develop a new analytic tool, this manuscript may not be interesting enough to the general readers of Nature Communications. The first part of the result section (2.1 Modeling metabolism using transcriptomes) cited several times of the "methods", but it is hard to follow this without knowing which part of the "methods". For example, "a list of metabolites available for uptake (Methods)" doesn't seem to exist in the current manuscript. These "methods" should be included as part of the "results" section, since this manuscript is about establishing a new computational framework METAFlex.

Reviewer #3 (Remarks to the Author): expertise in metabolomics bioinformatics

The study by Huang and colleagues describe and present a novel bioinformatic framework that uses bulk or single-cell RNAseq to infer metabolic fluxes in cancer. The authors use datasets from cell lines in culture, patient tumor data from the cancer genome atlas (TCGA), and scRNA-seq from different immunotherapy interventions to validate their approach.

Studying metabolic communication between different cell populations in the tumor microenvironment is a highly relevant question in the field. Their approach provides a tool that can be quite useful to address this question and their findings might be of general interest to the

cancer metabolism community. However, there are some points that the authors should address to strengthen their conclusions.

Major Comments:

1. One of the main caveats of the study is the lack of experimental validation. For example, the authors studied glucose uptake flux in LUSC and the LUAD tumors and predicted that LUSC tumors had higher glucose uptake flux than LUAD tumors. This prediction could be verified by ¹³C-glucose tracing in cell lines derived from both types of tumors.
2. The authors focused mainly on glucose uptake flux, however many cell types in the tumor microenvironment are highly dependent on other metabolic pathways (e.g. serine, arginine, polyamine metabolism and fatty acid oxidation). The study would greatly benefit if the authors use METAFlux to provide insight into these routes in different cell types of the tumor microenvironment.
3. A recent study (Wagner et al, 2021) described an algorithm to characterize metabolic states from single-cell RNA-Seq data. The authors refer to this study very briefly in the discussion. Although the methodology used is different compared to METAFlux, the study would solidify if the authors discuss on the performance of both approaches using scRNA-seq data

Minor comments.

1. In the paragraph starting at line 56, it is not clear if the authors refer to ¹³C metabolic flux or Seahorse measurements. In line 59 it would seem that the authors claim that OCR and ECAR can be measured mainly with ¹³C-MFA.

Reviewer #4 (Remarks to the Author): expertise in computational analysis of metabolomics data

Authors presented a computational framework called METAFlux that uses transcriptomic data to infer metabolic fluxes, validated the framework using experiments, TCGA and scRNA-seq data on diverse cancer. METAFlux can characterize metabolic reprogramming in the tumor microenvironment using transcriptomic data. In general, the article is relatively clear and informative, but there are some questions that need to be answered before deciding whether to publish it or not:

1. In the article, the authors mention the the Compass metabolic model recently developed by Wanger et al. What is the difference between METAFlux and Compass? What is the advantage of this algorithm compared to others?
2. Using predicted metabolic fluxes, authors identified a cluster of "LUSC-like LUAD" samples with upregulated glucose uptake and worse prognosis. This is an interesting phenomenon, is it possible to identify this group of patients with FDG-PET in clinical practice?
3. In the prognosis analysis, some more analyses with associations on clinical variables would be appreciated. What is the effect of treatment, T, N, grades ect.? A multivariate model would be helpful?
4. The metabolic characteristics of the four types of cells are analyzed in Fig4. Given the heterogeneity of immune cells (e.g., divided into T cells, B cells, NK cells, myeloid cells, etc.), is it possible to perform a more granular metabolic analysis of cell subpopulations.
5. In Figure 5, the authors found that the metabolic preferences of different cells. Further validation by experiments is necessary to illustrate the reliability of the algorithm. For example, the expression of key metabolic enzymes in various cells can be evaluated.
6. In the manuscript, the authors mention that "glucose uptake of the TME is well correlated with myeloid infiltration level in the lymph nodes", but as shown in Figure5C, the correlation is however not very high.
7. In the single-cell analysis of metastatic LUAD, non-tumor cells from different patient also showed some heterogeneous differences in the UMAP, which was not consistent with previous single-cell analyses, how can this be explained?

8. Some statistical tests need to be added to the figures, such as Fig5D, Fig 6D.

RESPONSE TO REVIEWERS' COMMENTS

Reviewer #1 (comments to the author): expertise in lung cancer metabolism

Here, Huang et al invented a novel computational framework, METAFlex (METAbolic Flux balance analysis) to infer metabolic fluxes in cancers from transcriptomic data. The authors included a workflow for bulk and single cell RNA seq data, and addressed the interaction of different cell types in the tumor microenvironment. Moreover, nutrient availability, as an important constraint for metabolic fluxes, was considered, however in a binary fashion (nutrient present or absent in plasma or in cell culture medium). The study is timely, metabolism prediction tools from expression data could be highly useful to the field, if their limitations would be taken into account. In the present study, the prediction of the relative contribution of different cell types in the TME to the uptake/release of different metabolites in lung adenocarcinoma is of interest and the time-dependent changes of metabolism in the CAR-NK cell treatment (immunotherapy) model provide novel insights. Although the results of the study certainly demonstrate the potential of this platform to analyze tumor metabolism and metabolic interactions between cancer cells and the TME, the model "validation" is largely based on application of the model to various datasets.

As a first approach the authors aimed to validate their model using the published dataset from the study on Human1, the genome-scale metabolic model which served as the basis for the present paper. The dataset comprised release or uptake data of metabolites of 11 cancer cell lines in vitro (ref 52). Uptake/release rates correlated with the predicted uptake values for the majority of metabolites analyzed, however, the model appears to show a weakness in predicting glucose uptake, which is a key feature in cancer cells and e.g. T cells. Moreover, the authors performed Seahorse measurements using Raji lymphoma cells co-cultured with NK cells, yet, a detailed methods description for the Seahorse assay is missing. As a weakness of the study, central metabolic pathways like glycolysis-branching pathways (e.g. the pentose phosphate pathway), amino acid, nucleotide or lipid synthesis pathways, or the TCA cycle were apparently not validated. This would require more elaborative assays, like stable isotopic tracing, still some effort could have been made in validating predictions for these pathways as well. Other possible approaches include essentiality prediction and experimental verification. As another limitation of the study, the real extracellular tumor microenvironment was not modeled. As the authors discuss, this is at present a difficult task, due to a lack of extracellular fluid metabolomics data for most tumors

Major points:

#1. Fig. 2a and b: Correlations of predicted uptake/release rates for 26 metabolites in 11 different cell lines from the NCI-60 panel is shown, with an overall positive Spearman correlation coefficient. However, glucose and lactate, highly important metabolites in cancer, show no or even a negative correlations between predicted and measured uptake/release rates, although the experiments are performed in defined conditions in vitro. The model might be even poorer in predicting glucose and lactate in vivo fluxes in complex environments. Still, the remainder of the paper highly focusses on glucose uptake. How did the authors validate that glucose uptake is sufficiently well predicted by the model, except for the comparison of key results with published data (e.g. higher Glucose uptake in lung squamous cell carcinoma compared to lung adenocarcinoma, and high uptake by myeloid cells, Fig 4)? Fig. 1c suggests that the model seemingly well predicts the direction of flux of most metabolites, however, when comparing different cells, treatment, or tissues also the extent of flux, not only the direction (uptake or release), are important.

Response: We acknowledge the limitation in using NCI-60 panel for examining glucose uptake and that the correlations for other metabolites are more consistent. We believe that the experimental model may not fully capture the complexity of glucose metabolism in a physiologically relevant context due to the difference in nutrient and metabolic requirements between cell lines grown in controlled environments and actual tumors in the tumor microenvironment. As such, we sought to validate METAFlex's ability to predict glucose uptake on an alternative dataset using human blood profile (**Supplemental Tables S6**), which is more representative of in vivo conditions.

As suggested by the reviewers, we further validated the predicted glucose uptake in cancer tissue samples using FDG-PET [1]. The Standardized Uptake Value (SUV) derived from FDG-PET, a measure of glucose uptake calculated from the radioactivity concentration in tissue and the injected dose of radioactivity per kilogram of the patient's weight, was used as a ground truth for comparison. A cohort of 84 triple-negative breast cancers (TNBCs) with independent gene expression data and matched SUV data was analyzed under human blood profile medium. The results showed a statistically significant Spearman correlation of 0.31 (P value = 0.003) between the predicted glucose uptake and the measured SUV (**Supplementary Tables S5**), indicating that the METAFlex method has the potential for predicting glucose uptake in larger cohorts and under more physiologically relevant conditions. We have updated this result at **lines 184 to 194** to reflect these additional analyses and their implications.

In addition, we have also applied METAFlex to a pancreatic cancer dataset and found similar accuracy in predicting FDG. For a detailed account of these results, please refer to the response to your Question #2.

#2. The approach by Huang et al to include physiological constraints rather than modelling metabolism based on fluxes derived from in vitro data alone is a potential advantage. Still, as the authors state in the Discussion, the model is based on the absence or presence of metabolites (binary variable) instead of including available metabolite concentrations, which is a limitation. The authors address this issue in the discussion. To address this issue, the paper could be improved by assessing METAFflux results in tumors showing a very low perfusion/nutrient abundance, e.g. pancreatic cancer, and to test the accuracy of the model prediction (as shown in Fig. 2b) e.g. using tumor cells or immune cells cultured in plasma-like medium or starvation medium . Does the model still provide meaningful results in such situations? The authors should explicitly mention this limitation also in the online version of METAFflux and in which situations this may lead to incorrect results. Moreover it should be made clear, for which cell culture medium the framework is designed, since they vary widely (e.g. DMEM medium does not contain certain amino acids, like aspartate or asparagine).

Response: In response to the reviewer's comment, we applied METAFflux on a dataset of pancreatic cancers (N=8) to assess its accuracy in predicting glucose uptake in tumors with low perfusion and nutrient abundance[2], using a human blood profile(**Supplemental Tables S6**) for physiologically relevant samples. The dataset contains matched gene expression and Standardized Uptake Values (SUV) derived from FDG-PET. We found that the predicted glucose uptake by METAFflux achieved a high Spearman correlation of 0.81 (P value= 0.02 with the SUVs (**Supplementary Tables S5**). These results demonstrate that METAFflux is able to accurately predict glucose uptake in pancreatic cancers, even in conditions of low perfusion and nutrient availability. Although we have not tested METAFflux under comprehensive starvation conditions, our current model based on the general human blood profile performs well in predicting glucose uptake for pancreatic cancers.

We would like to note that users are not restricted to the pre-defined metabolite list for general use purposes. We have provided the option to modify the input metabolite list if users have knowledge of their background metabolite profiles. This feature will allow for greater flexibility in adapting the model to various conditions, including starvation medium.

In the manuscript, we have clarified that while our current model shows promising results, it may be subject to limitations in specific situations. We encourage users to tailor the model based on their knowledge of the specific cancer type and metabolic environment. Regarding the cell culture medium, we have updated the list and detailed discussion in the manuscript (**lines 195 to 201 and Supplemental Table S5, S6, Methods 4.4.2, and Supplementary Methods 1**).

#3. A detailed description of this profile of “consumable” metabolites is missing. According to the methods (section 4.4.2.), the list of consumed or secreted metabolites contains 44 metabolites present in cell culture growth medium, which are not limited. The model does not

allow for uptake of the remaining metabolites, the ones that are limited, as is stated in methods 4.2.2. However, usually it is the rapid uptake of metabolites rendering them limiting in vitro, thus it appears unclear, why the model is designed in a manner not allowing the cells to uptake these “remaining” metabolites from the medium (see line 445). Glucose is a limiting factor according to Ref 52, so thus the model by Huang et al exclude glucose uptake? How is it then computed in the analyses in the rest of the paper?

Response: We thank the reviewer’s great comment. The profiles of the consumable list of metabolites are now added to the manuscript (**Supplemental Tables S6**). We also added additional information to clarify our method (**Methods 4.4.2**).

Previously we stated that the uptake or secretion rates of these 44 metabolites are not limited. For the remaining metabolites in the model, we do not allow cells to uptake them from the medium but instead, allow cells to secrete them into the medium. We meant to indicate that the uptake or secretion rates of those 44 metabolites are not limited under the “constraint” space at the modeling setup step (please refer to method section 4.4.3 equation 3), but eventually the uptake or secretion fluxes for 44 metabolites are constrained by gene expression and steady-state assumption at the model calculation step.

It is true that those rapidly uptaken metabolites are rate-limiting. However, we often do not have prior knowledge of what those metabolites are for different cases, thus defining a reasonably smaller list to feed into the model can be a good starting point. At this point, we only offered a binary list of significant metabolites present in the environment. The reason we need to define the metabolite availability is to guide the optimization search in a biologically relevant sub-space. We further showed in our control experiments that when cells or samples were allowed to uptake or secrete **all** metabolites, the optimization results were less desirable than when the pre-defined list of metabolites was made available.

Regarding glucose, glucose is part of the metabolite list for cell line medium and human blood medium. We did not exclude glucose from the metabolite list in our analysis presented in the manuscript. It is true that glucose is a limiting factor. However, we can still include glucose in the metabolite list. If glucose is truly limited, the final glucose rates of uptake or secretion should reflect this phenomenon as those fluxes are collectively constrained by gene expression and steady-state assumption. We have clarified these points in the revised manuscript in the results section **lines 195-201**.

#4. Hypoxia is another important modulator of cell metabolism and is frequently present in tumors. Single cell RNA seq may potentially allow to include hypoxia in the model, however, it is not clear, whether this has been done.

Response: Thank you for the reviewer's insightful comment regarding hypoxia. Hypoxia, a condition of limited oxygen supply, is indeed a highly relevant phenomenon in tumor and tumor microenvironments. In our previous manuscript, we did not differentiate oxygen level difference since our model is only able to model presence or absence of metabolite. We did not perform a comprehensive analysis to evaluate hypoxia, but we believe that METAFlex can capture hypoxia conditions from gene expression constraints. Metabolic readaptation under hypoxia is common, and usually, oxygen-deprived cells increase their uptake of glucose and favor lactate production [3]. To address the question, we have performed additional analysis.

Below is the summary of the analysis:

To examine whether our model can capture such an association, we performed additional analysis using hypoxia signature scores calculated on TCGA pan-cancer data in a previous study [4]. mRNA-based hypoxia scores were available for 676 tumors. The scores were calculated by assigning a +1 score to patients with mRNA abundance in the top 50% for each gene in the hypoxia signature and a -1 score to those in the bottom 50%. This process was repeated for every gene in the signature, generating a hypoxia score for each patient. [4, 5]. Higher scores indicate hypoxia in tumor and lower scores normoxia.

To validate METAFlex, we regressed METAFlex features against the hypoxia scores: $lm(\text{Hypoxia scores} \sim \text{Oxygen_Uptake} + \text{Glucose_uptake} + \text{Lactate_secretion})$. Our results (supplemental table) showed that oxygen uptake was negatively associated with hypoxia score (Estimate = -0.19, CI = -0.27 – -0.12, $p < 0.001$), while glucose uptake was positively associated with hypoxia score (Estimate = 0.32, CI = 0.25 – 0.40, $p < 0.001$). Lactate secretion was also positively associated with hypoxia score (Estimate = 0.09, CI = 0.02 – 0.17, $p = 0.014$). These results align with previously described hypoxia-mediated metabolic reprogramming and demonstrate METAFlex's capacity to accurately capture hypoxia-associated metabolic adaptations. We have included the updated results at **results section 2.5 (line 262-282)** and **Supplementary Table S2**.

#5. Some results are interpreted by the authors in a manner suggesting that the actual glucose uptake rates are known. E.g. line 221, results: "However, a significant difference [of glucose uptake] emerged in immune cells ($P=0.045$, 95% CI of effect size [-1.8626, -0.1579], two-sample t-test), indicating that immune cells may not be deprived of glucose in the TMEs of the LUSCs". Gene expression certainly may predict the metabolic phenotype, as has been shown in several studies, however, whether this applies to glucose-limited environments is less well understood. This should be pointed out.

Response: We appreciate the reviewer's concern. We have modified the description of the results to convey that we lacked knowledge of the ground truth, that the glucose uptake rates were based on predicted values, and that further study will be needed to fully evaluate the

METAFlux's prediction in conditions like a nutrient-starved medium. The reflected changes are now in the manuscript from **line 296 to line 298**. Results section **lines 195-201** now have discussed the use of METAFlux under the glucose-limited condition.

#6. Certain metabolites, which are part of competing metabolic pathways but also can be released (e.g. citrate) are included in the cell-cell interaction model. How did the authors estimate release of such central metabolic pathway intermediates based on gene expression? A description of the underlying assumptions should be given and an experimental validation of citrate (or succinate) exchange with the extracellular space (which is of high interest in the field) should be ideally done.

Response: We thank the reviewer's comment. In short, METAFlux uses a total of 3,625 metabolic-related gene expressions to calculate metabolic reaction activity scores which are further used as the constraints for flux balance analysis in step 4.4.3. The flux balance analysis can simultaneously estimate 13,082 reactions. The citrate uptake or release can be extracted by querying the predicted output called "HMR_9286", which describes the exchange of citrate. We have clarified the method of retrieving the flux in line **625-627**.

We acknowledge the importance of experimental validation of citrate or succinate exchange with the extracellular space. However, we were unable to find any datasets that contained matched gene expression and citrate or succinate exchange data to perform such validation. We also think that performing experimental validation to a specific metabolite goes beyond our scope, which is about developing a new method to enable systemic estimation of flux of thousands of metabolites from gene expression data (particularly single-cell data). We have experimentally validated some of the results and plan to work with research community to further validate others.

#8. Although metabolic pathways are complex and interconnected, the model may be flawed by including genes into a pathway that in fact mediate the opposite pathway. E.g. in Supplementary Table S2 ("Glycolytic pathway genes"), FBP1, FBP2 are included, which are not glycolytic but exclusively gluconeogenic (=reverse pathway). Also, G6PC, G6PC2 and G6PC3 do not mediate glycolysis, but gluconeogenesis or glycogenolysis. The authors should re-examine their pathway genesets manually for misannotations.

Response: We appreciate the reviewer's attention to potential misannotations in our gene set (Supplementary Table S4). We would like to clarify that METAFlux itself does not rely on pathway definitions; it uses flux balance analysis on the genome-scale metabolic model (GEM) of Human1. The pathway definitions were employed for benchmarking against gene set enrichment calculations, which do not account for directionality and interaction among cell types.

In our previous analysis, we computed the glucose uptake for all cell types using METAFlex and aimed to compare the results with popular gene set scoring methods like ssGSEA and Seurat AddModuleScore. These methods require pre-defined gene sets, and we initially used the genes from the 'glycolysis and gluconeogenesis' pathway, which was not an accurate representation of the glycolysis score.

To address the reviewer's concerns, we have updated our gene set to the Reactome gene set for the glycolysis pathway only (**Supplementary Table S4**), which will provide a more accurate estimation of the "glycolysis score". We emphasize that this change in gene set does not affect the METAFlex model, as it does not rely on pathway-level annotation to compute flux. The updated results are now reflected at the manuscript at **line 365-369**.

#9. Fig. 6c: are the differences significant? (Seahorse data)

Response: The p-value between the comparison of CAR19 and CAR19/IL15 is not significant, likely due to the low number of samples (sample size=3). We now additionally reported Hedges'g value to estimate the magnitude of the difference between groups. For basal respiration, the Hedges'g between CAR19/IL15 and CAR19 is 1.63 with 95% CI [-0.47,2.72]. For glycolysis, the Hedges'g between CAR19/IL15 and CAR19 is 2.30 with 95% CI [-0.04,4.62]. For both basal respiration and glycolysis, the Hedges'g estimates are considered large sizes. This suggests that the trend is robust and statistical significance was not achieved due to the limited sample size. Seahorse data statistical testing is now added in **line 427 to line 431** and **Figure 6C**.

Minor points:

1. Line 238: TME cells cannot "produce" phenylalanine, strictly speaking, since phenylalanine is an essential amino acid. They can just release it from their stores (e.g. after proteolysis).

Response: We have revised the text. Instead of stating that TME cells "produce" phenylalanine, we now clarify that TME cells can release phenylalanine. This change has been made in **line 317-318** now.

2. Suppl Table S1: Ions are not nutrients, thus the caption should be changed. Reformatting the table (Superscripts and Subscripts) is needed. Also oxygen is not a nutrient (Main text line 236).

Response: Instead of referring to the ions and oxygen as "nutrients," we have adopted the term "metabolites," which better captures their role in cellular metabolism. Furthermore, we have reformatted the table and figures in Figure 6A to ensure the proper use of superscripts and subscripts. The changes have been made at **line 314-318** and **supplementary table S3, Figure 6a**.

3. No method description for the Seahorse assay and for the Raji/NK cell co-culture experiment is given. How were the NK cells obtained?

Response: We have clarified these in our revised manuscript. The source manuscript, "Loss of metabolic fitness drives tumor resistance after CAR-NK cell therapy and can be overcome by cytokine engineering", is in press at Science Advance. Once the paper is online, we will revise the citation accordingly. In the meantime, we have provided a brief overview of the experiment below:

Seahorse assay: The extracellular acidification rate (ECAR) was measured using the Agilent Seahorse XFe96 Analyzer (Agilent) following the manufacturer's protocol as previously described (29). In Fig. 1G, NT, CAR19 and CAR19/IL-15 NK cells were assayed either alone or purified after 2 hours of co-culture with Raji targets followed by the Seahorse assay. Fig. S6A is an independent experiment where ECAR was measured by Seahorse assays using 2 g/L D-glucose, 2.5 μ M oligomycin and 100 mM 2-Deoxyglucose (2-DG) followed by live cell counting using Hoechst 33342 (Invitrogen) dye in Cytation 1 cell imaging multi-mode reader (Agilent).

Raji/NK experiment: The CD19⁺ Raji Burkitt lymphoma cell line (CCL-86), and the K562 erythroleukemia cell line (CRL-3344) were obtained from the American Type Culture Collection (Manassas, VA). K562 cells were retrovirally transduced to co-express 4-1BBL, CD48, and membrane-bound interleukin (IL)-21 and served as universal antigen presenting cells (uAPC) for in vitro NK cell expansion (30). Raji and K562 cells were cultured in RPMI-1640 (Invitrogen) supplemented with 10% fetal bovine serum (FBS; HyClone), 1% penicillin-streptomycin, and 1% GlutaMAX™. Peripheral blood samples used in this study were collected from two patients treated on a clinical trial of adoptive therapy with cord blood (CB)-derived NK cells expressing iC9/CAR19/IL-15 (CAR19/IL-15) at MD Anderson Cancer Center as previously reported (NCT03056339). All patients gave informed consent following Institutional Review Board (IRB)-approved protocols. All studies were performed in accordance with the Declaration of Helsinki. CD56⁺CD3⁻ NK cells isolated from cord blood units obtained from the MD Anderson Cancer Center Cord Blood Bank were purified using an NK negative isolation kit (Miltenyi Biotec), and co-cultured with irradiated (100 Gy) uAPCs at a 1:2 (NK cells: uAPC) ratio in complete stem cell growth medium (SCGM), supplemented with 200 U/ml recombinant human IL-2 (Proleukin). On Day 4 post uAPC stimulation, fresh NK cells were purified again and transduced with retroviral vectors expressing the CAR constructs described above.

Expanded NT or CAR-transduced NK cells (10×10^6 /mouse) were injected through the tail vein on day 0. We started with 13-15 mice per group for the single infusion in vivo experiment and 15 mice per group for the double infusion in vivo experiment. Five (5) mice per group were followed for survival and tumor cells were quantified weekly using Living Image software as previously described (15). Additionally, 8-10 mice were followed for single cell analysis with 2 mice per group sacrificed at 4 timepoints (days 7, 14, 21 and 28 for the NT and CAR19 groups) for the experiments presented in Fig. 2 or 5 timepoints (days 7, 14, 21, 28, and 35 and days 14, 21, 28, 35 and 61 for double infusion group for the CAR19/IL-15 group) for the experiments presented in Figs. 2 and 6. At each time point, we collected liver, spleen, blood and bone

marrow, and cells were harvested and cryopreserved for later analysis by scRNA-seq and mass cytometry.

4. Fig. 2b: How is biomass defined and why was this parameter included in metabolite uptake/release assays?

Response: We appreciate the question regarding the definition of biomass and its inclusion in the metabolite uptake/release assays. We would like to highlight that the manuscript has provided a detailed definition of biomass in the Discussion section, which refers to the flux of a generic biomass reaction (biomass_human from Human1). This reaction encompasses various cellular components necessary for cellular proliferation, including proteins, lipids, and small metabolites, among others, and has been proven to be a reliable characterization of the composition and demand of human cells. To further address the reviewer's query, we wish to emphasize that we have opted to exclude "biomass" from the uptake or release assay. This decision is based on the fact that "biomass" is a pseudo-reaction that differs from other real metabolites. The updated figures are **Figure 2b and 2c**.

5. Introduction, line 57 “¹³C metabolic flux analysis (13C-MFA) and Seahorse Extracellular Flux (XF) analyzer, have become widely used in metabolic research. 13C MFA is the current gold standard in measuring intracellular fluxes of central carbon metabolism and to assess cells' extracellular bioenergetic state. It can measure OCR (Oxygen consumption rate: an indicator of mitochondrial respiration) and ECAR (extracellular acidification rate”: The last sentence refers to the XF Analyzer, not to 13C-MFA. 13C-MFA can assess metabolic fluxes based on label enrichment in key metabolites following incubation with stable isotopic tracers (such as ¹³C-labeled precursors).

Response: We have revised the statement for clarity as (**line 58-60**): ¹³C-MFA is the current gold standard in measuring intracellular fluxes of central carbon metabolism, while the Seahorse Extracellular Flux (XF) analyzer is the benchmark for assessing cells' extracellular bioenergetic state. Although ¹³C-MFA offers valuable insights, it has limited usage in deriving large metabolic networks [6]. Seahorse Extracellular Flux (XF) analyzer can measure OCR (Oxygen consumption rate: an indicator of mitochondrial respiration) and ECAR (extracellular acidification rate: an indicator of glycolysis) of living cells simultaneously in real-time[7]. Even though the Seahorse platform provides valuable insight into the functional status of cells, fluxes of other metabolites are not measured.

6. Introduction: To complete the summary on existing approaches to assess metabolism in human or model tumors, *in vivo* stable isotopic labelling should be mentioned, not just PET (which PET?), see e.g. Faubert, Berardinis RD et al, doi: 10.1038/s41596-021-00605-2.

Response: We have updated the reference and content in manuscript **line 72 to line 76**. The changes are as following: For *in vivo* metabolic assessment, approaches such as positron

emission tomography (PET) are used. These include techniques like ^{18}F -fluorodeoxyglucose (FDG) PET and ^{18}F -Glutamine[8]. Additionally, in vivo stable isotope tracing has emerged as a novel approach, utilizing stable isotope-labeled nutrients to investigate metabolic activity within intact tumors[9]. Notably, these methods are limited to probing specific subsets of metabolic reactions[8].

7. Methods, 4.4.2. (line 442), Defining nutrient availability profile for cell culture and patient samples: It is not clear, which 44 metabolites were selected to be defined as “non-limiting” and why. In order to make the constraint model more clear, the authors should not just state “We use the cell line culture medium, containing 44 metabolites as the growth medium in cell line models (Ref 52)”, but provide a list of these 44 metabolites. Ref 52 reports on the use of Ham’s medium for their model, however which Ham medium was used (Ham F-10 or F-12) is not specified), also if serum was added to the medium.

Response: We have provided a detailed list of the 44 metabolites defined as "non-limiting" in our constraint-based model, which can be found in **supplemental table S6**, sheet ‘*cell line medium (Ham's)*’. The rationale behind selecting these metabolites was to reduce the optimization search space to a biologically relevant sub-space, as mentioned earlier in our response and added in the manuscript **line 580-586**.

Our 44-metabolite list was adapted from the Human1 study[10]. We have analyzed the profiles of both Ham's F-12 and Ham's F-10 nutrient mixes and highlighted the included metabolites in red in our supplemental xx Excel sheet ‘*Ham medium composition*’. While Ham's F-12 and Ham's F-10 share some similarities, there are differences in the composition and concentration of nutrients. However, our METAFlex input only considers the presence or absence of metabolites for initiating predictions, without taking concentration into account. To validate the appropriateness of those 44 metabolites, we examined the components of the Ham's F-12 and Ham's F-10 media, we only included metabolites involved in our metabolic network model. Furthermore, we added essential components like water and oxygen, cofactors, and inorganic phosphate. We also added fatty acids, vitamins that are typically found in serum and also present in our metabolic network, to account for the nutrients provided by the serum supplementation commonly used in cell culture to support cell growth. The updated description is now included in the **supplementary method section 1**.

For more general use purposes, users are not restricted to the pre-defined metabolite list. We have provided the option to modify the input metabolite list if users have knowledge of their background metabolite profiles. This description has been updated at **line 593-596**.

8. Ref 6 is an abstract. Given the large body of literature on the role of metabolic perturbations on cancer aggressiveness the authors may find different literature to fit to their introduction.

Response: We have replaced Ref. 6 with a journal paper: Altered propionate metabolism contributes to tumour progression and aggressiveness. The updated text is now at **line 39-41** as follows: “A recent study shows that dysregulated propionate metabolism increases the metastatic potential for breast and lung cancer[11].”

9. Fig. 3d and S2f: How is overall survival defined? This should be stated in “methods”. Patients at risk or total number of patients

Response: We have updated our survival analysis method and risk table accordingly. To evaluate the prognostic significance of the identified lung adenocarcinoma (LUAD) cancer clusters derived from either METAFlex or gene expression-based profiles, we performed a survival analysis using the TCGAanalyze_survival() function from the TCGAbiolinks package in R[12]. The overall survival refers to the time from diagnosis until death. Patients were stratified into groups based on their cluster assignments, and overall survival was analyzed for each group. The log-rank test was utilized to compare survival distributions across the groups, and p-values were reported to assess the statistical significance of the differences in overall survival between the clusters. Please see the **supplementary method section 6** survival analysis for updates.

10. Section 2.4. It should be “¹⁸F-FDG (fluorodesoxyglucose) PET, not “FGD-PET”.

Response: We have corrected the typo in our current manuscript. The updated content is now at **line 223-225**.

Reviewer #2 (comments to the author) expertise in metabolic flux analysis

This manuscript “Characterizing cancer metabolism from bulk and single-cell RNA-1 seq data using METAFlex” reported a new computational framework METAFlex (METAbolic Flux balance analysis), which can infer metabolic fluxes from bulk or single-cell transcriptomic data. The authors clearly summarized the current approaches for analyzing the metabolic data and their limitations. The introduction is well-written, and the authors pointed out the gaps in the current approaches. However, it is unclear how the current METAFlex is improved compared to the previous ones, such as ecGEMs, as figure 2c shows there are largely similar. Metabolically, this study highly focuses on glucose uptake (Fig2-5). Since glucose uptake can be directly measured, it is unclear if and how the current METAFlex can provide accurate information about the other metabolism. Together, although this is an interesting study, which is trying to develop a new analytic tool, this manuscript may not be interesting enough to the general readers of Nature Communications.

The first part of the result section (2.1 Modeling metabolism using transcriptomes) cited several times of the “methods”, but it is hard to follow this without knowing which part of the “methods”. For example, “a list of metabolites available for uptake (Methods)” doesn’t seem to exist in the current manuscript. These “methods” should be included as part of the “results” section, since this manuscript is about establishing a new computational framework METAFlex.

Response: We sincerely appreciate the reviewer's critical comments. Although Figure 2c displays a similar directional accuracy between METAFlex and ecGEMs, it is important to consider both direction and intensity correlation when predicting metabolic profiles. METAFlex demonstrates an improvement in directional accuracy for over seven metabolites and increased correlation for more than 15 metabolites, as seen in Figure 2b. Furthermore, all cell lines experience an increase in correlation. Notably, METAFlex has applications for both bulk and single-cell RNA-seq data. In the context of single-cell RNA-seq, it considers interactions between various cell types within the TME, allowing for a more in-depth exploration of the complex relationships and dependencies between different cell types and their metabolic processes in the context of cancer. This application is not considered by the ecGEMs model.

To showcase a broader range of metabolites, we have included other metabolites in the CAR-NK study (**Results section 2.8**), which includes oxygen, lactate, amino acids, etc. Additionally, we revised our manuscript to include more metabolites in the lung cancer single-cell study (**Results section 2.7**), such as serine, arginine, and carnitine (**from line 347 to line 364, and Supplementary Note**).

Furthermore, we have updated the metabolite medium file as **supplementary table S6** and updated the description at line **581-596**.

In response to the reviewer's concern regarding the "Methods" section, we have revised the manuscript to improve the overall flow and understanding for the reader. We revised the section titles and subtitles to more accurately reflect the content, which should allow readers to find the information more efficiently. For each citation of "Methods" in the main text, we

now provide a more specific reference, such as Method 4.2 to guide the readers directly to the corresponding section.

Reviewer #3 (Remarks to the Author): expertise in metabolomics bioinformatics

The study by Huang and colleagues describe and present a novel bioinformatic framework that uses bulk or single-cell RNAseq to infer metabolic fluxes in cancer. The authors use datasets from cell lines in culture, patient tumor data from the cancer genome atlas (TCGA), and scRNA-seq from different immunotherapy interventions to validate their approach.

Studying metabolic communication between different cell populations in the tumor microenvironment is a highly relevant question in the field. Their approach provides a tool that can be quite useful to address this question and their findings might be of general interest to the cancer metabolism community. However, there are some points that the authors should address to strengthen their conclusions.

Major Comments:

1. One of the main caveats of the study is the lack of experimental validation. For example, the authors studied glucose uptake flux in LUSC and the LUAD tumors and predicted that LUSC tumors had higher glucose uptake flux than LUAD tumors. This prediction could be verified by ¹³C-glucose tracing in cell lines derived from both types of tumors.

Response: We agree this suggestion. Unfortunately, we could not find any ¹³C-glucose tracing data from the lung cancer cell-lines. Fortunately, our results are supported by literature. In particular, the study by Leitner, B. P. et al. utilized ¹⁸F-FDG PET-CT scans data from open source TCIA (The Cancer Imaging Archive) to examine glucose uptake in LUSC and LUAD tumors[13]. They found that the SUVmax normalized to SUV in the descending aorta was one-fold higher in LUSC than in LUAD. This was observed despite comparable body weight, stage, adipose tissue, or muscle volume. Notably, the researchers found that this effect was specific to the tumor rather than a systemic effect. No differences were observed in ¹⁸F-FDG uptake in non-tumor tissues, including the heart, liver, skeletal muscle (deltoid), adipose tissue (SubQ abdominal), and brain. We emphasized this point in the revised manuscript (**Supplementary Note**).

We also performed additional validation using gene expression data with matched FDG-PET. Please refer to our responses to reviewer 1 #1 and #2, and to reviewer 4 #2 for more details.

2. The authors focused mainly on glucose uptake flux, however many cell types in the tumor microenvironment are highly dependent on other metabolic pathways (e.g. serine, arginine, , polyamine metabolism and fatty acid oxidation). The study would greatly benefit if the authors use METAflux to provide insight into these routes in different cell types of the tumor microenvironment.

Response: We appreciate the suggestions. We have now added analysis including additional metabolites, as shown in the figure and the paragraph below:

Serine is a crucial precursor for protein synthesis, nucleic acids, and lipids. Upregulated serine metabolism has been correlated with increased cell proliferation and poor prognosis in various tumors[14]. The differential metabolic uptake and release patterns of serine observed in our study highlight the complex interplay between tumor cells and the surrounding microenvironment in metastatic LUAD. The release of serine by B and T cells may provide an essential nutrient source for epithelial (tumor) cells, contributing to their growth and proliferation. Moreover, serine uptake by epithelial(tumor) cells could further support tumor cell growth by facilitating the generation of other essential metabolites and promoting nucleotide synthesis. The observed serine uptake by myeloid cells suggests that myeloid cells may also rely on serine as a nutrient source, potentially competing with tumor cells for this essential amino acid. This competition relationship suggests that myeloid cells could indirectly modulate serine availability to tumor cells, impacting their metabolic and proliferative capabilities. For Arginine, epithelial (tumor) cells displayed the highest arginine uptake. This elevated arginine uptake in tumor cells suggests an increased demand for protein synthesis, polyamine biosynthesis, and nitric oxide production. A previous study suggests tumor relies heavily on environment for arginine when undergoing catabolic stress[15]. Ornithine, another polyamine precursor, was uptaken by all cell types. The elevated ornithine uptake by B cells might reflect the increased demand for polyamines of B cells. Carnitine plays a vital role in the transfer of long-chain fatty acids across the inner mitochondrial membrane for beta-oxidation[16]. All cell types exhibit an uptake of carnitine, and myeloid cells displayed the highest uptake of carnitine followed by epithelial (tumor) cells. This observation suggests a higher reliance on fatty acid metabolism for epithelial (tumor) and myeloid cells.

The updated manuscript **line 347-364** now reflects the changes. The corresponding figure is updated as **Figure 5.e** and **supplementary figure 4a**.

3. A recent study (Wagner et al, 2021) described an algorithm to characterize metabolic states from single-cell RNA-Seq data. The authors refer to this study very briefly in the discussion. Although the methodology used is different compared to METAflux, the study would solidify if the authors discuss on the performance of both approaches using scRNA-seq data.

Response: We agree that discussing the performance of both approaches using scRNA-seq data could help solidify our study. However, we would like to clarify that several fundamental differences have made it non-informative to perform a direct performance comparison between the two methods.

First, METAFlex utilizes the Human1 metabolic mode to perform the FBA, while Compass employs Recon2. These different underlying metabolic models result in unique reaction IDs, reaction counts, and scopes. Therefore, comparing the METAFlex with Compass would not provide an informative assessment of their relative merits, since the performance differences could be attributed to the underlying metabolic models rather than the algorithms themselves.

Second, METAFlex is specifically designed for analyzing metabolic profiles in the tumor context, capturing heterogeneity in the tumor microenvironment (TME), and elucidating metabolic interactions among different cell types by modeling all cell types as a single community. In contrast, Compass is a more general approach without constraints on the cellular community. Because it computes maximal fluxes for metabolic reactions, assigns penalties based on mRNA expression, and determines flux distributions that minimize overall penalties while maintaining a specified flux threshold.

Third, METAFlex requires bootstrapping for single-cell data, resulting in cluster-wise metabolic statistics. It provides insights onto the collective metabolic behavior, focusing on the metabolic landscape and interactions among various cell types. In comparison, Compass directly outputs single-cell-wise fluxes. This difference in resolution also makes it difficult to perform direct comparison.

Nonetheless, we have included a **Supplementary Note** in the revised manuscript to clarify these differences and discuss why their results are not directly comparable.

Minor comments.

1. In the paragraph starting at line 56, it is not clear if the authors refer to ^{13}C metabolic flux or Seahorse measurements. In line 59 it would seem that the authors claim that OCR and ECAR can be measured mainly with ^{13}C -MFA.

Response: We have revised the statement for better clarity as (**lines 58-60**): ^{13}C -MFA is the current gold standard in measuring intracellular fluxes of central carbon metabolism, while the Seahorse Extracellular Flux (XF) analyzer is the benchmark for assessing cells' extracellular bioenergetic state. Although ^{13}C -MFA offers valuable insights, it has limited usage in deriving large metabolic networks [6]. Seahorse Extracellular Flux (XF) analyzer can measure OCR (Oxygen consumption rate: an indicator of mitochondrial respiration) and ECAR (extracellular acidification rate: an indicator of glycolysis) of living cells simultaneously in real-time[7]. Even though the Seahorse platform provides valuable insight into the functional status of cells, fluxes of other metabolites are not measured.

Reviewer #4 (Remarks to the Author): expertise in computational analysis of metabolomics data

Authors presented a computational framework called METAFlex that uses transcriptomic data to infer metabolic fluxes, validated the framework using experiments, TCGA and scRNA-seq data on diverse cancer. METAFlex can characterize metabolic reprogramming in the tumor microenvironment using transcriptomic data. In general, the article is relatively clear and informative, but there are some questions that need to be answered before deciding whether to publish it or not:

1. In the article, the authors mention the Compass metabolic model recently developed by Wanger et al. What is the difference between METAFlex and Compass? What is the advantage of this algorithm compared to others?

Response: We appreciate the reviewer's insightful comment. We have included in the **Supplementary Note** a detailed discussion about the differences between METAFlex and Compass. Also, please refer to our response to reviewer 2 Question#3. In conclusion, METAFlex provides several advantages over Compass.

Firstly, METAFlex focuses on providing a comprehensive understanding of the metabolism within the TME, as our underlying model Human1 demonstrates significant improvement over other GEMs[10]. Thus, it allows for a more accurate representation of the metabolic processes within the TME, providing a deeper understanding of tumor cell metabolism and its complexities. This enables researchers to investigate the metabolic processes and pathways specific to cancer.

Second, METAFlex considers the interactions between various cell types within the TME, allowing for a more in-depth exploration of the complex relationships and dependencies that exist between different cell types and their metabolic processes in the context of cancer. This aspect is not considered by Compass.

2. Using predicted metabolic fluxes, authors identified a cluster of "LUSC-like LUAD" samples with upregulated glucose uptake and worse prognosis. This is an interesting phenomenon, is it possible to identify this group of patients with FDG-PET in clinical practice?

Response: We appreciate the reviewer's insightful comment regarding identifying the "LUSC-like LUAD" patient group using FDG-PET in clinical practice. To investigate the possibility, we obtained the matched TCGA LUAD FDG-PET metadata containing Standardized Uptake Value (SUV) max values (peak FDG uptake for the highest voxel within the tumor) from a previous study[13]. The overlapping FDG-PET patient cohorts contain four LUAD1 and nine LUSC-like LUAD patients.

Consistent with our METAFflux analysis, we found that LUSC-like LUAD patients exhibited higher glucose uptake, as indicated by a p-value of 0.086. Although the p-value is not strictly significant due to small sample size, the observed trend is consistent with our analysis. We have described these new results in line 246-250 and the supplementary figure 2d.

3. In the prognosis analysis, some more analyses with associations on clinical variables would be appreciated. What is the effect of treatment, T, N, grades ect.? A multivariate model would be helpful?

Response: We appreciate the feedback regarding our prognosis analysis of overall survival and agree that additional analyses with associations on clinical variables would be valuable.

To address this, we performed a multivariate analysis using Cox regression on clinical data from TCGA LUAD cancer patients. Our model included clinical variables such as the AJCC TNM staging, previous treatment information, age, race, gender, and our METAFflux clusters (LUAD1 and LUSC-like LUAD) [supplement table].

Our results showed that the cluster [LUAD1] remained significantly associated with reduced mortality risk compared to reference group LUSC-like LUAD even after adjusting for these clinical variables.

Specifically, the HR estimate for the cluster [LUAD1] was 0.64 with a 95% CI of 0.44 – 0.93 and a p-value of 0.019. The presence of ajcc pathologic n [N1] and ajcc pathologic n [N2] are also significant predictors of mortality compared to the baseline group N0, with HRs of 2.07 (95% CI: 1.44 – 2.97; p < 0.001) and 2.61 (95% CI: 1.75 – 3.90; p < 0.001), respectively. On the other hand, the presence of ajcc pathologic n [N3] and ajcc pathologic n [NX] are not significantly associated with any increased or decreased risk of mortality. Among the other variables, prior

treatment (Yes) is also a significant predictor of mortality, with an HR of 11.65 (95% CI: 3.44 – 39.41; $p < 0.001$) compared to the prior treatment (No). The presence of ajcc pathologic t [T3] and ajcc pathologic t [T4] are also significant predictors of mortality compared to baseline T0, with HRs of 3.15 (95% CI: 1.70 – 5.83; $p < 0.001$) and 2.47 (95% CI: 1.14 – 5.34; $p = 0.022$), respectively. Other factors like age, gender, and race are not associated with mortality.

Overall, these findings indicate that cluster [LUAD1] remains a robust prognosis predictor of survival in LUAD patients. We had included the new results at line 242-244 and Supplementary Table S1.

4. The metabolic characteristics of the four types of cells are analyzed in Fig4. Given the heterogeneity of immune cells (e.g., divided into T cells, B cells, NK cells, myeloid cells, etc.), is it possible to perform a more granular metabolic analysis of cell subpopulations.

Response: Yes, it is possible on the single-cell data but difficult on the bulk-sorted data. As suggested, we have carried out additional subpopulation analysis on the single-cell lung cancer data directly. We identified and analyzed metabolic characteristics for various immune cell subpopulations.

For glucose, the subpopulation analysis revealed that all cell types exhibit an uptake of glucose, which suggests glucose competition exists in LUAD. The heterogeneous landscape of glucose uptake profiles highlights the complex interplay within the tumor microenvironment. Among all cell types, monocyte-derived macrophages (mo-Macs) and monocytes exhibited elevated glucose uptake, followed by MALT B cells (mucosa-associated lymphoid tissue B cells). In contrast, Follicular B cells and GC B cells in the DZ (germinal center (GC) B cells in the dark zones) showed lower glucose uptake. This heterogeneity in glucose uptake among B cell subsets reflected differences in their metabolic states within the LUAD microenvironment.

For oxygen, similarly, all cell types exhibited an uptake of oxygen, which suggests oxygen competition exists in LUAD. Mo-Mac cells showed the highest oxygen uptake, followed closely by CD207+CD1a+ LCs and epithelial cells (tumor) cells. This suggests that these cell types rely more on oxidative phosphorylation for energy production. Moreover, data revealed a distinct

release and uptake pattern of asparagine and isoleucine, underscoring the cooperative metabolic behavior among various cell types in LUAD. For example, most immune cell subsets, including cytotoxic CD8+ T, exhausted CD8+ T, CD8 low T, regulatory T (Treg), CD207+CD1a+ Langerhans cells (LCs), CD4+ T helper (Th), monocytes, MALT B cells, naive CD4+ T, and naive CD8+ T, primarily release asparagine. This release pattern implies that these immune cells may supply asparagine to the asparagine-consuming cells within the tumor microenvironment, thus fostering metabolic cooperation.

We have included these results in the **Supplementary Note** and reflect the changes in manuscript **line 363-364**.

Note: CD1c+ DCs (Langerhans cells, LCs), monocyte-derived macrophages (mo-Macs). GC B cells in the DZ (germinal center (GC) B cells in the dark zones which undergo rapid proliferation and somatic hypermutation), MALT B cells (MALT b cells mucosa-associated lymphoid tissue B cells).

5. In Figure 5, the authors found that the metabolic preferences of different cells. Further validation by experiments is necessary to illustrate the reliability of the algorithm. For example, the expression of key metabolic enzymes in various cells can be evaluated.

Response: We appreciate the concern and agree that experimental validation would indeed provide additional support for our findings. However, we would like to point out that we do not have the biological replicates for the analyzed samples and conducting cell line functional validation might not be informative in this context since we believe there might be a metabolic difference between cell line and patient samples. Nonetheless, we recognize the importance of validation. Therefore, we have analyzed the expression of key metabolic genes from the original dataset and from an independent dataset. The expression of these key metabolic genes is already encoded in the predicted fluxes derived from our algorithm. Previous research suggested that both *GLUT1 (SLC2A1)* and *GLUT3 (SLC2A3)* have positive correlations with FDG-PET SUV values in NSCLCs (non-small cell lung cancers)[17].

Thus, we extracted the expression of glucose transporter genes *SLC2A1* and *SLC2A3* from the original dataset and calculated the module scores for glucose uptake using the Seurat `AddModuleScore` function.

The key gene signatures scores showed a consistent trend of glucose uptake predictions made by METAFflux, with the order being Myeloid cells > B cells > T cells > Epithelial cells. This consistency between gene expression profiles and our algorithm's predictions provides further support for the reliability of our model.

The independent dataset (GSE1239020) comprising brain (n = 3), bone (n = 1), and adrenal (n = 1) lung adenocarcinoma metastases[18], we extracted the expression of glucose transporter genes *SLC2A1* and *SLC2A3* and calculated the module scores. Our analysis revealed a similar trend to our findings in metastatic lymph node samples, with myeloid cells exhibiting the highest glucose transporter scores and tumor cells having the lowest scores. The order between T cells and B cells is changed, which is likely attributable to sampling differences. Nevertheless, this observation consistently shows the highest glucose uptake in myeloid cells and the lowest glucose uptake in tumor cells, supporting the robustness of our finding. We have included a **Supplementary Note** in the revised manuscript to reflect the changes.

6. In the manuscript, the authors mention that “glucose uptake of the TME is well correlated with myeloid infiltration level in the lymph nodes”, but as shown in Figure5C, the correlation is however not very high.

Response: We thank the feedback. We acknowledge that the correlation between glucose uptake in the TME and myeloid infiltration level in the lymph nodes is indeed moderate, with a correlation coefficient of 0.36. We have revised our result interpretation to “glucose uptake of TME has a moderate correlation with myeloid infiltration level in the lymph nodes”. This moderate correlation still delivers a meaningful insight as numerous factors in TME may interact non-linearly. Furthermore, our finding agrees with previous studies that have reported a correlation between myeloid infiltration and FDG avidity [19, 20], reinforcing the biological relevance of results from METAFIux. We have incorporated these findings into our revised manuscript at line **338-340**.

7. In the single-cell analysis of metastatic LUAD, non-tumor cells from different patient also showed some heterogeneous differences in the UMAP, which was not consistent with previous single-cell analyses, how can this be explained?

Response: We appreciate the concern. We believe this was due to technical differences and does not affect the validity of our results. Allow us to clarify:

We selected 7 patients with mLN (patients with lymph node metastasis samples) for our study. The observed discrepancy can be attributed to differences in data preprocessing steps, such as different variable gene selection methods and the number of principal components used. In addition, it is worth noting that the authors of the previous single-cell analysis used the Seurat V2 package and employed tSNE projection instead of UMAP. While in our study, we used Seurat V4 and utilized UMAP as our projection method, which could further contribute to the observed differences. Furthermore, based on the plots of the previous single-cell study, even though each immune cell type forms a distinctive cluster, we can still observe some heterogeneous differences between different patients within each immune cell cluster. Both studies don't correct for batch effect, thus our results are not confounded by the batch effect. Finally, our METAFflux method does not rely on clustering or projection, we simply use the normalized metabolic gene expression. Thus, this difference does not affect the validity of our prediction results.

Figure 5A

Previous single-cell tSNE projection plot by sample and by cell type.

8. Some statistical tests need to be added to the figures, such as Fig5D, Fig 6D.

Response: We appreciate the suggestion. The statistical results are now added to **Figure 5d** and **Figure 6d**.

Reference

1. Kim, S.K., et al., *Genomic Signature of the Standardized Uptake Value in (18)F-Fluorodeoxyglucose Positron Emission Tomography in Breast Cancer*. *Cancers (Basel)*, 2020. **12**(2).
2. Crespo-Jara, A., et al., *A novel genomic signature predicting FDG uptake in diverse metastatic tumors*. *EJNMMI Res*, 2018. **8**(1): p. 4.
3. Lee, P., N.S. Chandel, and M.C. Simon, *Cellular adaptation to hypoxia through hypoxia inducible factors and beyond*. *Nat Rev Mol Cell Biol*, 2020. **21**(5): p. 268-283.
4. Bhandari, V., et al., *Divergent mutational processes distinguish hypoxic and normoxic tumours*. *Nat Commun*, 2020. **11**(1): p. 737.
5. Winter, S.C., et al., *Relation of a hypoxia metagene derived from head and neck cancer to prognosis of multiple cancers*. *Cancer Res*, 2007. **67**(7): p. 3441-9.
6. Saldida, J., et al., *Unbiased metabolic flux inference through combined thermodynamic and ¹³C flux analysis*. *bioRxiv*, 2020: p. 2020.06.29.177063.
7. Little, A.C., et al., *High-content fluorescence imaging with the metabolic flux assay reveals insights into mitochondrial properties and functions*. *Commun Biol*, 2020. **3**(1): p. 271.
8. DeBerardinis, R.J. and K.R. Keshari, *Metabolic analysis as a driver for discovery, diagnosis, and therapy*. *Cell*, 2022. **185**(15): p. 2678-2689.
9. Faubert, B., et al., *Stable isotope tracing to assess tumor metabolism in vivo*. *Nat Protoc*, 2021. **16**(11): p. 5123-5145.
10. Robinson, J.L., et al., *An atlas of human metabolism*. *Sci Signal*, 2020. **13**(624).
11. Gomes, A.P., et al., *Altered propionate metabolism contributes to tumour progression and aggressiveness*. *Nat Metab*, 2022. **4**(4): p. 435-443.
12. Colaprico, A., et al., *TCGAbiolinks: an R/Bioconductor package for integrative analysis of TCGA data*. *Nucleic Acids Res*, 2016. **44**(8): p. e71.
13. Leitner, B.P., et al., *Multimodal analysis suggests differential immuno-metabolic crosstalk in lung squamous cell carcinoma and adenocarcinoma*. *NPJ Precis Oncol*, 2022. **6**(1): p. 8.
14. Amelio, I., et al., *Serine and glycine metabolism in cancer*. *Trends Biochem Sci*, 2014. **39**(4): p. 191-8.
15. Al-Koussa, H., et al., *Arginine deprivation: a potential therapeutic for cancer cell metastasis? A review*. *Cancer Cell Int*, 2020. **20**: p. 150.
16. Longo, N., M. Frigeni, and M. Pasquali, *Carnitine transport and fatty acid oxidation*. *Biochim Biophys Acta*, 2016. **1863**(10): p. 2422-35.
17. Meyer, H.J., A. Wienke, and A. Surov, *Associations between GLUT expression and SUV values derived from FDG-PET in different tumors-A systematic review and meta analysis*. *PLoS One*, 2019. **14**(6): p. e0217781.
18. Laughney, A.M., et al., *Regenerative lineages and immune-mediated pruning in lung cancer metastasis*. *Nat Med*, 2020. **26**(2): p. 259-269.

19. Mabuchi, S., et al., *Pretreatment tumor-related leukocytosis misleads positron emission tomography-computed tomography during lymph node staging in gynecological malignancies*. *Nat Commun*, 2020. **11**(1): p. 1364.
20. Reinfeld, B.I., et al., *Cell-programmed nutrient partitioning in the tumour microenvironment*. *Nature*, 2021. **593**(7858): p. 282-288.

REVIEWERS' COMMENTS

Reviewer #1 (Remarks to the Author):

Response to response to point #1. The authors performed a number of additional analyses in additional datasets to test their model and it appears that the data substantiated the performance of the model for glucose uptake. The respective results are shown in Table S5. However, XY graphs would be easier to read and should be included. Still, as in version 1 of the manuscript, glucose and lactate fluxes (and others like methionine) show negative correlations of measured and predicted fluxes in the cell line panel (Fig. 2b). If the model works well for glucose fluxes in vivo, but fails to predict certain metabolite fluxes in vitro in certain settings, this limitation and possible explanations need to be discussed in the discussion.

Response to response to point #3. The authors provided the option to modify the input metabolite list if users have knowledge of their background metabolite profiles. This improvement of the model reflects recent developments in the field addressing metabolism heterogeneity in response to different nutrient availabilities and is highly appreciated.

Response to response to Minor point #3 (No method description for the Seahorse assay and for the Raji/NK cell co-culture experiment is given. How were the NK cells obtained? Response: We have clarified these in our revised manuscript. The source manuscript, "Loss of metabolic fitness drives tumor resistance after CAR-NK cell therapy and can be overcome by cytokine engineering", is in press at Science Advance. Once the paper is online, we will revise the citation accordingly. In the meantime, we have provided a brief overview of the experiment below.)

Thank you for summarizing the methods. However, in the revised version there is still no method description or citation. Please include the method summary and citation in the text (e.g. "Co-culture of [...] and Seahorse assay [...] were performed as described in....").

The authors addressed all other points satisfactorily.

Points to address:

#1 The correlations of data in Table S5 should additionally be shown as plots.

#2. Negative correlations for measured and predicted glucose fluxes in vitro (Fig. 2b) should be discussed.

#3. Please include the co-culture and Seahorse method summary and citation in the text (e.g. "Co-culture of [...] and Seahorse assay [...] were performed as described in....").

#4. The new hypoxia dataset is not properly referenced. In line 269 "Ref [4]" is indicated, which is a review. This is obviously a mistake that occurred due to copying the rebuttal letter text, where Ref 4, Bhandari V et al, is cited.

Additional small points:

#1. Line 164, Results: The sentence "We selected 11 cell lines for evaluation, as other cell lines had nutrient depletion ...[...]" is confusing and should be revised. Why did those cells have nutrient depletion? Because of missing metabolites in the medium (e.g. Aspartate is missing from DMEM) or due to rapid consumption or any other cause? A cell line as such does not show a priori nutrient depletion.

#2. Line 302: The significantly enhanced glucose signature in LUSC immune cells and LUAD immune cells is explained by different glucose availabilities ("[...]..indicating that immune cells may not be deprived of glucose in the TMEs of the LUSCs compared with those in the LUADs.") However, different compositions and activation states of immune cells may also play a role (has this been checked?). The authors may wish to include these considerations.

- #3. Fig. 2b "decreasing concentration" instead of "decreases concentration"
- #4. Fig. 4bcd: Are the data for LUAD or LUSC? This should be mentioned in the legend.
- #5. Fig. 6d: It would improve readability if the definition of the competition score (PCCS) would be included in the legend.
- #6. Fig. 6b and all other (also Supplementary) Figures: sub and superscript characters should be checked (e.g. here H^+ instead of H^+).
- #7. In Table S1 control groups (group used for HR calculation) should be indicated (e.g. "v.s. N0").
- #8. Table S2: the caption "The linear regression model coefficients for Hypoxia_scores ~Oxygen_uptake+Glucose_uptake+Lactate" should be revised to make it more clear.
- #9. Table S6: Reference number or doi should be given instead stating "Retrieved from Human1 Publication".
- #10. Figures S1, S2 and S5a: inconsistent and small size of letters.
- #11. Fig S3a legend: the number of patients should be stated.
- #12. Methods line 515: "Golgi apparatus" instead of "golgi apparatus"
- #13. Methods line 591: "Cantor et al" instead of "Jason et al"

Reviewer #2 (Remarks to the Author):

This revision didn't fully address my previous concerns. For example, results 2.8 didn't provide much more information on intracellular metabolism, which is also raised by reviewer 1. Result 2.7 showed a few selected metabolites, but it is unclear how and why these metabolites are selected. Again, this is an interesting study, which is trying to develop a new analytic tool, this manuscript may not be interesting enough to the general readers of Nature Communications.

Reviewer #3 (Remarks to the Author):

The authors have addressed satisfactorily all my concerns. I recommend publication of the corrected manuscript in Nature Communications

Reviewer #4 (Remarks to the Author):

The author addressed all my problems.

RESPONSE TO REVIEWERS' COMMENTS

Reviewer #1 (Remarks to the Author):

Response to response to point #1. The authors performed a number of additional analyses in additional datasets to test their model and it appears that the data substantiated the performance of the model for glucose uptake. The respective results are shown in Table S5. However, XY graphs would be easier to read and should be included. Still, as in version 1 of the manuscript, glucose and lactate fluxes (and others like methionine) show negative correlations of measured and predicted fluxes in the cell line panel (Fig. 2b). If the model works well for glucose fluxes in vivo, but fails to predict certain metabolite fluxes in vitro in certain settings, this limitation and possible explanations need to be discussed in the discussion.

We have now included this in the discussion section and also addressed in the #2 below. In line 427-432 of the Discussions section, we have now added:

“Although the strengths of METAFlex have been successfully demonstrated, some limitations remain. First, we found that glucose uptake didn't correlate as consistently as other metabolites in NCI-60 panel. This discrepancy potentially results from the limitations of the cell-line models which do not accurately encapsulate the multifaceted nature of glucose metabolism. Alternatively, it suggests that the methodology implemented by METAFlex need improvement to fully capture the intricacies of metabolic processes under certain scenarios.”

Response to response to point #3. The authors provided the option to modify the input metabolite list if users have knowledge of their background metabolite profiles. This improvement of the model reflects recent developments in the field addressing metabolism heterogeneity in response to different nutrient availabilities and is highly appreciated.

Response to response to Minor point #3 (No method description for the Seahorse assay and for the Raji/NK cell co-culture experiment is given. How were the NK cells obtained? Response: We have clarified these in our revised manuscript. The source manuscript, "Loss of metabolic fitness drives tumor resistance after CAR-NK cell therapy and can be overcome by cytokine engineering", is in press at Science Advance. Once the paper is online, we will revise the citation accordingly. In the meantime, we have provided a brief overview of the experiment below.)

Thank you for summarizing the methods. However, in the revised version there is still no method description or citation. Please include the method summary and citation in the text (e.g. “Co-culture of [...] and Seahorse assay [...] were performed as described in....”).

This point has been addressed in #3 below.

The authors addressed all other points satisfactorily.

Points to address:

#1 The correlations of data in Table S5 should additionally be shown as plots.

We have added the data as plots in supplementary figures 1a and 1b.

#2. Negative correlations for measured and predicted glucose fluxes in vitro (Fig. 2b) should be discussed.

We have added the description of negative glucose correlation in line 170-174: In our study, we recognize the constraints of employing the NCI-60 panel for examining glucose uptake, given that correlations for other metabolites appear more consistent. This discrepancy may be because the experimental model may not fully capture the complexity of glucose metabolism, and there is a difference in nutrient and metabolic requirements between cell lines in controlled environments and tumors in the tumor microenvironment.

#3. Please include the co-culture and Seahorse method summary and citation in the text (e.g. “Co-culture of [...] and Seahorse assay [...] were performed as described in....”).

We have now added in line 363-366: In the Raji/NK experiment, NK cells were purified, transduced, and injected into mice for in vivo experiments, with survival and tumor cell quantification conducted. More detailed protocols are available in our previous work¹. More details regarding the Seahorse assay protocol can be found in our previous study¹.

#4. The new hypoxia dataset is not properly referenced. In line 269 “Ref [4]” is indicated, which is a review. This is obviously a mistake that occurred due to copying the rebuttal letter text, where Ref 4, Bhandari V et al, is cited.

The correct references have been updated.

Additional small points:

#1. Line 164, Results: The sentence “We selected 11 cell lines for evaluation, as other cell lines had nutrient depletion ...[...]” is confusing and should be revised. Why did those cells have nutrient depletion? Because of missing metabolites in the medium (e.g. Aspartate is missing from DMEM) or due to rapid consumption or any other cause? A cell line as such does not show a priori nutrient depletion.

We have revised the manuscript to add the reference behind our choice of cell lines, as mentioned in line 152-153. In order to ensure comparable results, we selected cell lines that consistently exhibit reliable metabolic profiles following the work done by Robinson et al². Our decision was further influenced by the study conducted by Nilsson et al., which highlighted that metabolic depletion could be a consequence of insufficient culture medium volumes or prolonged culture durations³.

#2. Line 302: The significantly enhanced glucose signature in LUSC immune cells and LUAD immune cells is explained by different glucose availabilities (“[..].indicating that immune cells may not be deprived of glucose in the TMEs of the LUSCs compared with those in the LUADs.”) However, different compositions and activation states of immune cells may also play a role (has this been checked?). The authors may wish to include these considerations.

We have now added additional discussion at line 284-287: indicating that immune cells may not be deprived of glucose in the TMEs of the LUSCs compared with those in the LUADs. Alternatively, it can also suggest that a specific subtype of immune cells may drive glucose uptake in a population of immune cells. Moreover, the metabolic profiles of immune cells, which typically elevate upon activation, could be significantly influenced by variations in their activation states.

#3. Fig. 2b “decreasing concentration” instead of “decreases concentration”
The annotation has been updated.

#4. Fig. 4bcd: Are the data for LUAD or LUSC? This should be mentioned in the legend.
The data is for the average profile of LUAD and LUSC, which has been added in the legend.

#5. Fig. 6d: It would improve readability if the definition of the competition score (PCCS) would be included in the legend.
The definition of PCCS has now been added to the legend of figure 6d as the following: Per cell competition score (PCCS) between cancer and NK cells in TME from day 7 to day 28 for oxygen and glucose uptake, respectively for CAR19/IL15. Test: T-test. PCCS is defined as the ratio of the per cell nutrient uptake flux in the tumor cells over that in the NK cells.

#6. Fig. 6b and all other (also Supplementary) Figures: sub and superscript characters should be checked (e.g. here H⁺ instead of H+).
The sub/super script for figure 6a.b and supplementary figure S5.b have been corrected.

#7. In Table S1 control groups (group used for HR calculation) should be indicated (e.g. “v.s. N0”).
The reference level for calculating HR have been added to the caption of Table S1 as the following Reference group for HR calculation: Cluster [LUSC-like LUAD], ajcc pathologic n [N0], ajcc pathologic m [M0], ajcc pathologic t [T1], gender [female], prior treatment [No], race [american indian or alaska native].

#8. Table S2: the caption “The linear regression model coefficients for Hypoxia_scores ~Oxygen_uptake+Glucose_uptake+Lactate” should be revised to make it more clear.
The caption has been updated to make it more readable.

#9. Table S6: Reference number or doi should be given instead stating “Retrieved from Human1 Publication”.
The reference has been updated.

#10. Figures S1, S2 and S5a: inconsistent and small size of letters.
Those figures have now been updated in the supplementary figures file.

#11. Fig S3a legend: the number of patients should be stated.
The number of samples has been added to the plot.

#12. Methods line 515: “Golgi apparatus” instead of “golgi apparatus”
It has been corrected as Golgi apparatus.

#13. Methods line 591: “Cantor et al” instead of “Jason et al”
The reference has been updated.

Reviewer #2 (Remarks to the Author):

This revision didn't fully address my previous concerns. For example, results 2.8 didn't provide much more information on intracellular metabolism, which is also raised by reviewer 1. Result 2.7 showed a few selected metabolites, but it is unclear how and why these metabolites are selected. Again, this is an interesting study, which is trying to develop a new analytic tool, this manuscript may not be interesting enough to the general readers of Nature Communications. We appreciate the acknowledgement and suggestions on expanding the scope. In this study, we purposefully emphasized extracellular metabolite uptake and release because that can lead to comparatively fewer metabolites, which is important to achieve robust performance. We selected several metabolites for demonstration because they are important and of great interest to research community. Our study aimed to shed light on the uptake and release dynamics of those metabolites, an often-overlooked area. Moreover, our focus was driven by our interest to understand the interaction between cells and their environment, which is highly relevant in tumor microenvironments. Our manuscript already has a relatively large scope, which includes a new method (applicable to two modalities: bulk and single-cell data) and extensive results (8 sections). We will consider the reviewer's suggestions in a future study.

Reviewer #3 (Remarks to the Author):

The authors have addressed satisfactorily all my concerns. I recommend publication of the corrected manuscript in Nature Communications

Reviewer #4 (Remarks to the Author):

The author addressed all my problems.

- 1 Li, L. *et al.* Loss of metabolic fitness drives tumor resistance after CAR-NK cell therapy and can be overcome by cytokine engineering. *Science Advances*, In Press. (2022).
- 2 Robinson, J. L. *et al.* An atlas of human metabolism. *Sci Signal* **13**, doi:10.1126/scisignal.aaz1482 (2020).
- 3 Nilsson, A., Haanstra, J. R., Teusink, B. & Nielsen, J. Metabolite Depletion Affects Flux Profiling of Cell Lines. *Trends Biochem Sci* **43**, 395-397, doi:10.1016/j.tibs.2018.03.009 (2018).